## REPORT

# Near millimolar concentration of nucleosomes in mitotic chromosomes from late prometaphase into anaphase

Fernanda Cisneros-Soberanis[1]*, Eva L. Simpson[2]*, Alison J. Beckett[3], Nina Pucekova[1], Samuel Corless[1], Natalia Y. Kochanova[1], Ian A. Prior[3], Daniel G. Booth[2], and William C. Earnshaw[1]

**Chromosome compaction is a key feature of mitosis and critical for accurate chromosome segregation. However, a precise quantitative analysis of chromosome geometry during mitotic progression is lacking. Here, we use volume electron microscopy to map, with nanometer precision, chromosomes from prometaphase through telophase in human RPE1 cells. During prometaphase, chromosomes acquire a smoother surface, their arms shorten, and the primary centromeric constriction is formed. The chromatin is progressively compacted, ultimately reaching a remarkable nucleosome concentration of over 750 µM in late prometaphase that remains relatively constant during metaphase and early anaphase. Surprisingly, chromosomes then increase their volume in late anaphase prior to deposition of the nuclear envelope. The plateau of total chromosome volume from late prometaphase through early anaphase described here is consistent with proposals that the final stages of chromatin condensation in mitosis involve a limit density, such as might be expected for a process involving phase separation.**

## Introduction

Ever since mitosis was first described by Flemming in 1882 (Paweletz, 2001; Paulson et al., 2021; Flemming, 1882), the fundamental question of how chromosomes form their compact cylindrical structure has remained elusive. A pioneering study using electron microscopy to characterize the different mitotic stages revealed that the chromosomes start to condense close to the nuclear envelope and continue to develop their mature cylindrical structure after its dissolution (Robbins and Gonatas, 1964). However, that study lacked the technology to reconstruct complete cells in three dimensions.

Subsequently, light microscopy has largely dominated the analysis of mitotic chromosome structure during mitotic progression. In one study, the total chromatin volume was measured from $G_2$ to telophase using live cell imaging of NRK epithelial cells expressing photoactivatable GFP-tagged H2B (H2B–PAGFP) (Mora-Bermudez et al., 2007). During mitotic entry, the chromatin volume decreased approximately two to threefold. The authors then followed a defined segment of single chromosomes using photoactivation during chromosome congression and subsequent mitotic exit. They concluded that

maximal chromosome compaction occurs in late anaphase due to an axial shortening of the chromosome arms. However, the volume of a cylinder increases linearly with length but with the square of the radius, and the width of the chromatids was not reported.

Another study, using FLIM-FRET to measure chromosome compaction in HeLa cells stably constitutively expressing H2B-EGFP/mCherry-H2B, reported a step-wise decrease of the mean EGFP fluorescence lifetime value between prometaphase and anaphase B (Lleres et al., 2009). This suggested an increase in chromosome condensation in anaphase B followed by a rapid decompaction at telophase onset. Both studies noted limitations, including lower resolution on the Z-axis and difficulty distinguishing between anaphase stages based on images of the fluorescently labeled chromosomes alone.

A more recent study building on pioneering work from the Laemmli lab (Adolph et al., 1977a, 1977b; Paulson and Laemmli, 1977; Marsden and Laemmli, 1979; Earnshaw and Laemmli, 1983) characterized chromosome compaction in early mitosis using a multidisciplinary approach involving high-resolution

[1]Wellcome Centre for Cell Biology, University of Edinburgh, Edinburgh, UK; [2]Biodiscovery Institute, University of Nottingham, Nottingham, UK; [3]Molecular Physiology and Cell Signaling, Institute of Systems, Molecular and Integrative Biology, University of Liverpool, Liverpool, UK.

Correspondence to William C. Earnshaw: bill.earnshaw@ed.ac.uk; Fernanda Cisneros-Soberanis: f.cisneros@ed.ac.uk; Daniel G. Booth: daniel.booth@nottingham.ac.uk

*F. Cisneros-Soberanis and E.L. Simpson are co-first authors.



microscopy, DNA–DNA interaction analysis, and polymer modeling. The study confirmed Laemmli's model that compact chromosomes form by reorganizing the chromatin using scaffold proteins and forming loop arrays (Marsden and Laemmli, 1979; Gibcus et al., 2018). Simulations of prometaphase chromosomes using coarse-grained models showed the presence of a 400-kb outer loop array formed by condensin II, within which was nested an array of ~80-kb loops formed by condensin I.

Recently, Serial Block Face Scanning Electron Microscopy (SBF-SEM) has enabled the comprehensive three-dimensional analysis of complete mitotic cells with high resolution. One prior study examining the chromosome structure of one prophase and one metaphase cell by correlative light and electron microscopy (CLEM) concluded that much of the volume of metaphase chromosomes comprises non-chromatin material (Booth et al., 2016). A second analysis used SBF-SEM to reconstruct and identify each chromosome based on size and centromere position in a prophase/prometaphase B-lymphocyte cell synchronized with colcemid. Confirming earlier results from chromosome-painting FISH studies (Croft et al., 1999; Cremer et al., 2003), it was concluded that chromosomes are distributed within the nucleus according to their gene density: with gene-rich chromosomes in the interior and gene-poor chromosomes localized near the nuclear periphery (Sajid et al., 2021).

Here, we analyze changes in chromosome structure during mitotic progression from prometaphase through telophase in 14 human RPE1 cells. We evaluate changes in chromosome structure including surface granularity, width, length, and volume of cohesed chromosomes and individual chromatids. Our determination of total chromosome volume reveals a remarkably high concentration of nucleosomes that remains nearly constant from late prometaphase through mid-anaphase.

## Results and discussion
### Synchronous mitotic entry in RPE1 hTERT CDK1as cells
We used chemical genetics to obtain a clone of RPE1 hTERT cells capable of undergoing synchronous mitotic entry regulated by an analog-sensitive (CDK1[as]) cDNA from *Xenopus laevis* sensitive to the ATP analog 1NM-PP1 (Fig. S1 A) (Gibcus et al., 2018; Hochegger et al., 2007). Following a 20 h incubation in 1NM-PP1, cells were in $G_2$ and entered mitosis after washout of the drug. However, 50% of these cells assembled multipolar spindles and underwent apoptosis shortly after mitotic exit (Fig. S2).

To reduce the time required in $G_2$ arrest, we incubated the culture with the CDK4/6 inhibitor palbociclib and then released the cells in fresh media prior to the addition of 1NM-PP1 (Trotter and Hagan, 2020). This enabled us to achieve efficient synchronous mitotic entry with a much shorter incubation in 1NM-PP1 (palbociclib -24 h; release -12 h; 1NM-PP1 8 h—release -> mitotic entry). Using this protocol, around 80% of the population was in mitosis at 90 min after 1NM-PP1 washout, and by 150 min, most had re-entered interphase with only ~12% apoptotic cells (Fig. S1 B and Fig. S2 C).

The RPE1 hTERT CDK1[as] cells started to enter prophase by 30 min after 1NM-PP1 washout and reached prometaphase after 50 min (Fig. S1 D). Thus, these cells provide a powerful system with a synchronous population of cells freely traversing mitosis (e.g., without nocodazole, etc.). This allowed us to easily select brief mitotic stages that are normally extremely rare (e.g., early anaphase) for CLEM.

Our initial three-dimensional SBF-SEM analysis of mitotic cells (see next section) revealed that, surprisingly, our CDK1[as] clone had 47 chromosomes. This was confirmed in chromosome spreads. This aneuploidy did not arise during our targeting of the CDK1 gene, as the parental cell line (RPE1 hTERT) had a similar karyotype in nearly half of the cells (Fig. S1 E). Thus, the extra chromosome had been stably acquired before the genetic modification of the cell line.

Karyotype analysis of our cell lines revealed that all clones had an extra chromosome 12 and an insertion on chromosome X (Fig. S1 F). This altered karyotype does not affect our subsequent analysis, which focused on chromosomes 1–5 and chromosomes 19–22. These chromosomes can be identified unambiguously by their relative size and centromere position. This analysis confirmed that our three-dimensional SBF-SEM imaging could reliably resolve individual chromosomes and identify an unsuspected aneuploidy.

### Metaphase chromosomes have a highly reproducible volume
We obtained the 3D structure of entire mitotic cells by SBF-SEM with a sampling resolution of 4 nm in X and Y and a resolution in Z of 60 nm (corresponding to the microtome section thickness in the 3View). SBF-SEM allows accurate quantification of chromosome volume since the imaging does not depend on the calibration of a point spread function or analysis of features below the working resolution of the imaging system.

Our samples for SBF-SEM microscopy were prepared by conventional procedures involving fixation and embedding. This introduces the potential for some shrinkage, although, in a previous study, we found chromosome shrinkage to be minimal (Booth et al., 2016). Since our experiments were conducted by CLEM, we could measure the diameter of the same individual cells before and after embedding. We found that the preparation method causes a ~7% shrinkage of the cell diameter (a ~20% reduction in volume). Whether the chromosomes undergo similar shrinkage is not known, although our previous results suggest that they do not. It will be informative in future studies to perform a similar analysis on samples imaged by cryoelectron microscopy.

We first examined metaphase cells. Metaphase was chosen because this is the most clearly defined mitotic stage during which chromosome morphology remains relatively constant.

Metaphase cells selected for analysis using light microscopy were subsequently fixed, embedded in plastic, and processed for CLEM. Next, the plastic containing each cell was excised, trimmed, and mounted on a pin for scanning electron microscopy. Datasets were collected in a Quanta 250 scanning electron microscope (FEI) fitted with a 3View system (Gatan). Following sectioning/imaging of the entire cell, the resulting data set was analyzed using the program Amira (Thermo Fisher Scientific). Each image dataset has a full resolution size of 10–15 GB, so it was necessary to bin the data for computational analysis.

To determine how to reduce the image file size without compromising the structural information, we varied the voxel size (i.e., image binning) in Amira, while retaining the same image density threshold. Chromosome volume was relatively insensitive to voxel size, but the surface area changed dramatically depending on the image binning (Fig. S3). After assessing the performance of Amira, we selected a voxel size of 25 × 25 × 60 nm for all subsequent analysis.

To obtain accurate volume measurements in our 3D reconstructions, we had to accurately determine the borders of the chromosomes in the images (segmentation). We segmented chromosomes using a threshold defined by the difference in grayscale intensity values between the cytoplasm and chromosomes (Fig. 1 A). The accuracy of this approach is supported by the reproducibility of the volume of four independent metaphases (140 ± 4.8 µm³; Fig. 1 B). In a previous study, we found the volume of a single human metaphase plate to be 175.9 µm³ (Booth et al., 2016). This ~20% difference in volume may be due to the combination of improved sampling resolution of a new detector in the 3View system (previously 24 nm in X and Y, currently 4 nm in X and Y), image binning in the previous study (see Fig. S3), the use of the RPE1 hTERT cell clone in those experiments, the precise mitotic stage, and differences in the uranyl acetate staining.

The resolution of the SBF-SEM datasets allowed us to unambiguously identify chromosomes 1–5 and 19–22 based on their relative size and centromere position (Fig. 1, C and D). Chromosomes 1 and 3 are large metacentric chromosomes. Chromosome 2 is slightly submetacentric, while chromosomes 4 and 5 are submetacentric with much shorter p arms. Chromosomes 19 and 20 are small metacentrics while chromosomes 21 and 22 are small acrocentrics (Fig. 1 D).

Identification of those chromosomes in situ in cells allowed us to calculate several parameters by correlating the chromosome length and volume with the known DNA content (Table S1). Thus, measuring the lengths of the nine identified chromosomes in metaphase cells and taking into account the known DNA content of each yielded a packing ratio of 66 ± 5.7 Mb/µm (Fig. 1 F). This corresponds to an overall DNA density of 84.3 ± 2.86 Mb/µm³ (Fig. 1, B and E). Dividing the entire chromosome complement volume of 140 ± 4.8 µm³ by the predicted total DNA content of our RPE1 clone based on the T2T sequence (Nurk et al., 2022) yielded a similar value of 89 ± 3.04 Mb/µm³. Thus, this analysis suggests that all chromosomes reach a similar limit density in metaphase.

A recent publication suggested, based on light microscopy, that chromosome width correlates with the length of individual chromosomes (Kakui et al., 2022). We therefore determined the width of individual chromatids as well as the width of the entire chromosome (i.e., two sister chromatids) when the two sisters were tightly cohesed. In measuring these parameters, we avoided the centromeres and telomeres, which have a specialized three-dimensional organization.

Overall, the average width of a single chromatid was 0.68 ± 0.04 µm (n = 10 measurements per chromosome), while the average width of a cohesed chromosome was 1.23 ± 0.1 µm (Fig. 1 G). These values are consistent with a previous publication from our group analyzing a single metaphase cell (Booth et al., 2016). Notably, the chromosome width is about 10% less than the sum of the widths of the two sister chromatids. This suggests three possibilities. First, sister chromatids may be compressed along their interface with one another. Second, there may be a roughly 0.06 µm overlap (mixing) of the two sister chromatids. Third, chromatin could be "squeezed" out of the plane of pairing (i.e., pushed above or below the plane of contact of the two sisters). Our data argue against possibility 3, as when viewed in cross-section the sister chromatids display a circular profile, not an ellipsoid as model 3 would suggest.

Consistent with the report that shorter chromosomes are thinner than longer ones (Kakui et al., 2022), the chromatid width of chromosomes 19–22 was slightly thinner than that of chromosomes 1–5, and this difference was statistically significant (0.66 ± 0.04 versus 0.69 ± 0.04 µm, P = 0.004—Fig. 1 G, bottom). When we evaluated the full chromosome width for the same chromosomes, we again observed that chromosomes 19–22 were slightly thinner. However, this difference was not statistically significant (Fig. 1 G, top).

To further extend these observations, we correlated the chromatid/chromosome width versus the chromatid/chromosome length. Overall our data support a weak correlation between chromatid width and length, but this was less pronounced than reported previously based on light microscopy (Fig. S4) (Kakui et al., 2022).

## Chromosomes in early prometaphase are longer and have a more granular surface

We turned next to prometaphase, a cell cycle stage in which chromosomes undergo dynamic changes as the nuclear envelope breaks down and condensin I associates with the chromatin. We successfully segmented the chromosomes of two early prometaphases and two late prometaphases (Fig. 2 A). Chromosomes undergo a progressive condensation as prometaphase progresses. Thus, these chromosomes had a much more variable volume than their metaphase counterparts, with a volume reduction of 42% between early and later prometaphase.

To establish a metric to analyze progression through prometaphase, we staged prometaphase cells based on the degree of centrosome separation in each cell (Tanenbaum and Medema, 2010). When combined with the extent of scattering of the chromosomes, this allowed us to reliably distinguish between early and late prometaphase independent of chromosome morphology.

During prometaphase, the total chromosome volume decreased from 256 to 209 µm³ in early prometaphase to 184 µm³ and then 150 µm³ in later prometaphase (Fig. 2 A and Table S1). The latter approached the average volume of metaphase chromosomes (140 µm³—Fig. 1 B). In the earliest prometaphases, we could readily segment individual chromosomes, but could not unambiguously identify them because the centromeric constriction was not fully formed, and the position of the centromere was not clear. This was evident when we displayed the largest chromosome in each cell (Fig. 2 C).

As prometaphase progressed, the chromosome volume and surface area decreased (Fig. 2 D). The volume for the largest

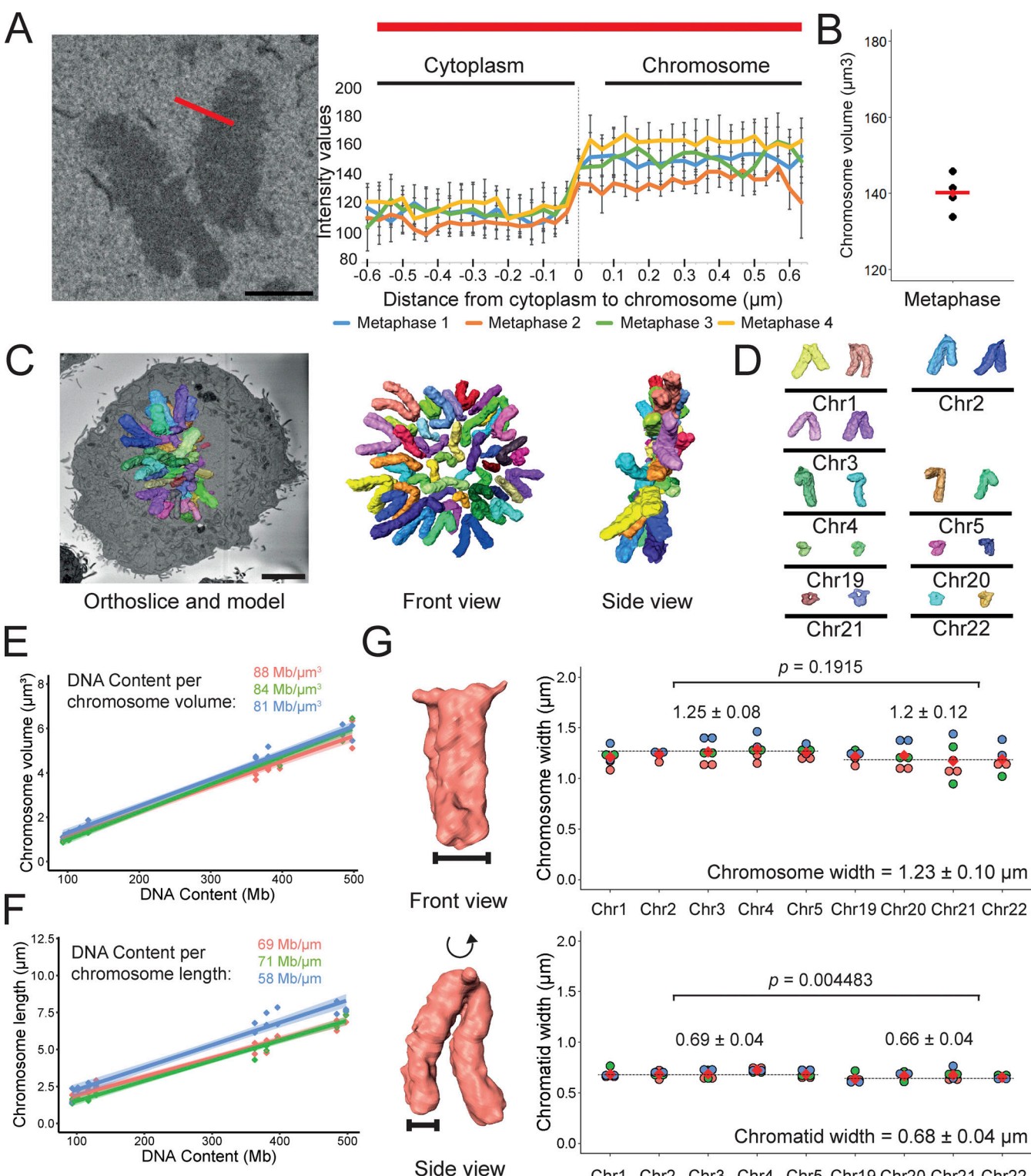

Figure 1. **Quantitative 3D electron microscopy shows that metaphase chromosomes have a reproducible volume. (A)** Representative image of an electron microscopy section of a mitotic RPE1 CDK1[as] (left) and the intensity grey-scale values between the cytoplasm and the chromosomes (right). The red line represents the analyzed region of the mitotic cell. Scale bar = 2 μm. **(B)** Volume quantification of four metaphases. The red line represents the average between the four cells analyzed. **(C)** Orthoslice and segmented model of chromosomes (left; scale bar = 5 μm), front view (middle), and side view (right) of a mitotic RPE1 CDK1[as] cell. **(D)** 3D karyotype and identification of chromosomes 1–5 and chromosomes 19–22. **(E)** Plot of chromosome volume versus DNA content used to calculate the DNA density in Mb/μm³ (n = 3). **(F)** Plot of chromosome length versus DNA content used to calculate the amount of DNA per μm (n = 3). **(G)** Cohesed chromosome (top) and chromatid (bottom) width. Left: Representation of chromosome 1 and the measurement taken. Right: Quantifications of chromosome and chromatid width ordered by chromosome in order of decreasing size (Chr1–5 and Chr 19–22). An average of 10 measurements per chromosome was plotted. The average chromosome width and SD for all the chromosomes are summarized below the graph. The average width ± SD is represented above the graph for large and small chromosomes. Two-tailed Student's t test, n = 3 cells, P = 0.01915 (chromosomes), P = 0.004483 (chromatids).

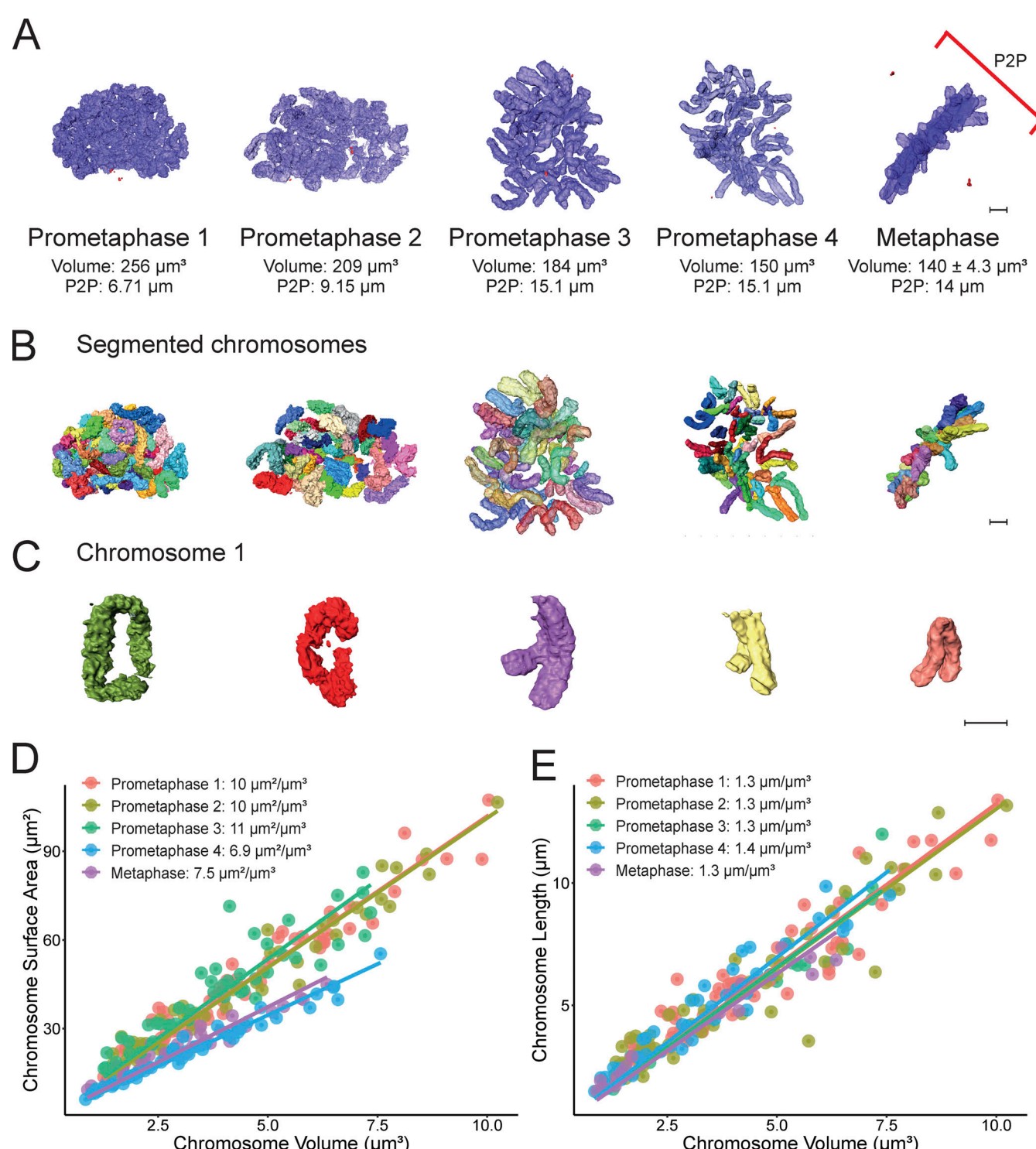

Figure 2. **Chromosomes in early prometaphase are longer, have a more granular surface, and lack a primary constriction. (A)** Representation of early (Prometaphase 1–2) and late (Prometaphase 3–4) prometaphase chromosomes and a metaphase chromosome complement for comparison. Total chromosome volume and pole-to-pole (P2P) distance (red line) are annotated underneath each image. Centrosomes are red. Scale bar = 2 μm. **(B)** Individual segmented chromosomes for the above models are identified by different colors. Scale bar = 2 μm. **(C)** Representation of the largest chromosome (Chromosome 1) for each of the above cells. Scale bar = 2 μm. **(D)** Correlation between chromosome volume and surface area of individual chromosomes. **(E)** Correlation between chromosome volume and length of individual chromosomes.

chromosome decreased by 37% from 10.03 μm³ in the earliest prometaphase to 6.35 μm³ in metaphase. This was accompanied by a decrease in the granularity of the chromosome surface, measured as a 58% reduction in surface area from 107.28 to 44.95 μm² in metaphase. Changes in surface granularity probably reflect deposition of the chromosome periphery compartment, which is ongoing during mid and late prometaphase (Booth et al., 2014, 2016; Cuylen et al., 2016; Stenström et al., 2020).

The length of the largest chromosome also decreased by almost 50% between the earliest prometaphase and metaphase (from 13.4 to 7.3 μm). Indeed, the shortening of the chromosomes appeared to be an ongoing gradual process during prometaphase (Fig. 2 E). Chromosome volume and length changes during prometaphase are likely to be driven at least in part by condensin I, which associates with the DNA after nuclear envelope breakdown (Gibcus et al., 2018; Hirota et al., 2004; Samejima et al., 2022). Surprisingly, analysis of the relationship between the total chromosome length and total chromosome volume revealed a relatively constant ratio of 1.3 ± 0.04 μm/μm³. Thus, chromatin compaction appears to increase in proportion to chromatid shortening during early mitosis.

### Chromatid length and width vary as anaphase progresses

To characterize changes in chromosome morphology during mitotic exit, we used CLEM to select four anaphase and two telophase cells for imaging and 3D reconstruction. Surprisingly, the first two anaphases were remarkably similar in volume to metaphase chromosomes (Anaphase 1: 145 μm³, Anaphase 2: 142 μm³ compared with an average total chromosome volume of 140 μm³ in metaphase—Fig. 3 A). However, total chromosome volume then increased as anaphase progressed (Anaphase 3: 164 μm³, Anaphase 4: 196 μm³). These volume changes did not correlate with increases in the granularity (i.e., surface area) of the chromosomes (Fig. 3 E).

A step-change increase in the total chromosome volume occurred once the nuclear envelope began to be deposited on the surface of the chromosomes (this defines the beginning of telophase): Telophase 1: 228 μm³, Telophase 2: 497 μm³ (Fig. 3 A). Indeed, the chromosomes in telophase 2 have nearly twice the volume of those in the earliest prometaphase.

Segmentation of the separated chromatids in anaphase cells (Fig. 3 A) allowed us to determine the DNA packing density for the RPE1 genome in early and late anaphase. In the earliest anaphases (1 and 2), the chromatids had DNA densities of 86.1 and 87.9 Mb/μm³, respectively. This is equivalent to the metaphase density of 89.2 Mb/μm³. Anaphase 3 was slightly more advanced with a slightly decreased density of 76.1 Mb/μm³. Anaphase 4 had a DNA density of 63.7 Mb/μm³ (Fig. 3 C and Table S1). Thus, the chromatin packing density undergoes a progressive decline during anaphase as the volume increases.

It was relatively straightforward to identify the separated chromatids for chromosomes 1–5 and 19–22 in anaphase cells (Fig. 3 B). During early anaphase, chromatids of these nine chromosomes had a linear DNA packing density similar to metaphase chromosomes. In metaphase, this figure is 66 Mb/μm, corresponding to 33 Mb/μm per sister chromatid. In early

anaphases (1 and 2), the DNA density remained remarkably conserved at 31 and 32 Mb/μm. As anaphase progressed, this linear density dropped slightly to 26 Mb/μm for Anaphase 3 and 28 Mb/μm for Anaphase 4 (Fig. 3 D). Thus, contrary to conclusions from light microscopy, the linear chromatin packing density along the chromosome arms does not vary during the earliest stages of anaphase chromatid separation (Nagasaka et al., 2016).

How can the overall packing density of anaphase chromatin decline while maintaining a relatively constant linear density? This occurs by increasing the diameter of the sister chromatids in Anaphase 4 (Fig. 3, F and G; and Table S1). Anaphase 3 appears to be an outlier with much longer and thinner chromatids. Given the proximity of the telomeres in this cell, we hypothesize that the chromatids are stretched, likely due to persistent catenation at or near telomeres.

Although it was difficult to resolve individual chromatid arms in Telophase, where this was possible, we observed a roughly 30% increase in chromatid width to 1.06 ± 0.17 μm (Fig. 3 F). This correlates with the sharp increase in the total chromosome volume during telophase.

Following segmentation and identification of the nine chromatids in anaphase, it was possible to determine whether there was a difference in chromatid width between large and small chromosomes. Indeed, there was a significant change in average chromatid width between large (Chr 1–5) and small (Chr 19–22) chromosomes (0.71 ± 0.08 versus 0.65 ± 0.08 μm; P = 0.005—Fig. 3 G and Fig. S4). This could either be due to differences in chromatin organization between longer and shorter chromosomes (Kakui et al., 2022) or to variable spindle forces since shorter chromosomes tend to be found in the spindle interior where the microtubule density (and spindle forces) are higher (Mosgöller et al., 1991; Chong et al., 2023, Preprint).

### Different spindle location and poleward movement of small and large chromosomes

In our 3D reconstructions here, we could identify and segment both chromosomes and structures required for their segregation, including kinetochores, microtubules, and centrosomes. This allowed us to compare the behavior of different groups of chromosomes during anaphase A and anaphase B. We measured the pole-to-pole distance (P2P) between centrosomes and found an increase from 14 μm at Metaphase to 21 μm in Anaphase 4 (Fig. 4 A). This increase in centrosome separation is consistent with published data (Su et al., 2016).

To characterize the behavior of a single chromosome in detail, we measured the distance between sister kinetochores (sK2K) and telomeres (sT2T) for the separated sister chromatids of chromosome 1 (Fig. 4 B). This analysis revealed that anaphase begins with a rapid separation between chromatids. The sK2K distance almost doubled from metaphase to early anaphase (Anaphase 1; 1.41 ± 0.05–3.11 ± 0.18 μm) while the sT2T distance increased from 0.61 ± 0.09 to 0.98 ± 0.14 μm. These increases were not accompanied by any detectable separation of the poles, suggesting that they correspond to anaphase A (Desai et al., 1998).

In the slightly more advanced Anaphase 2, the sK2K distance doubled again to 7.01 ± 0.40 μm but the sT2T distance for

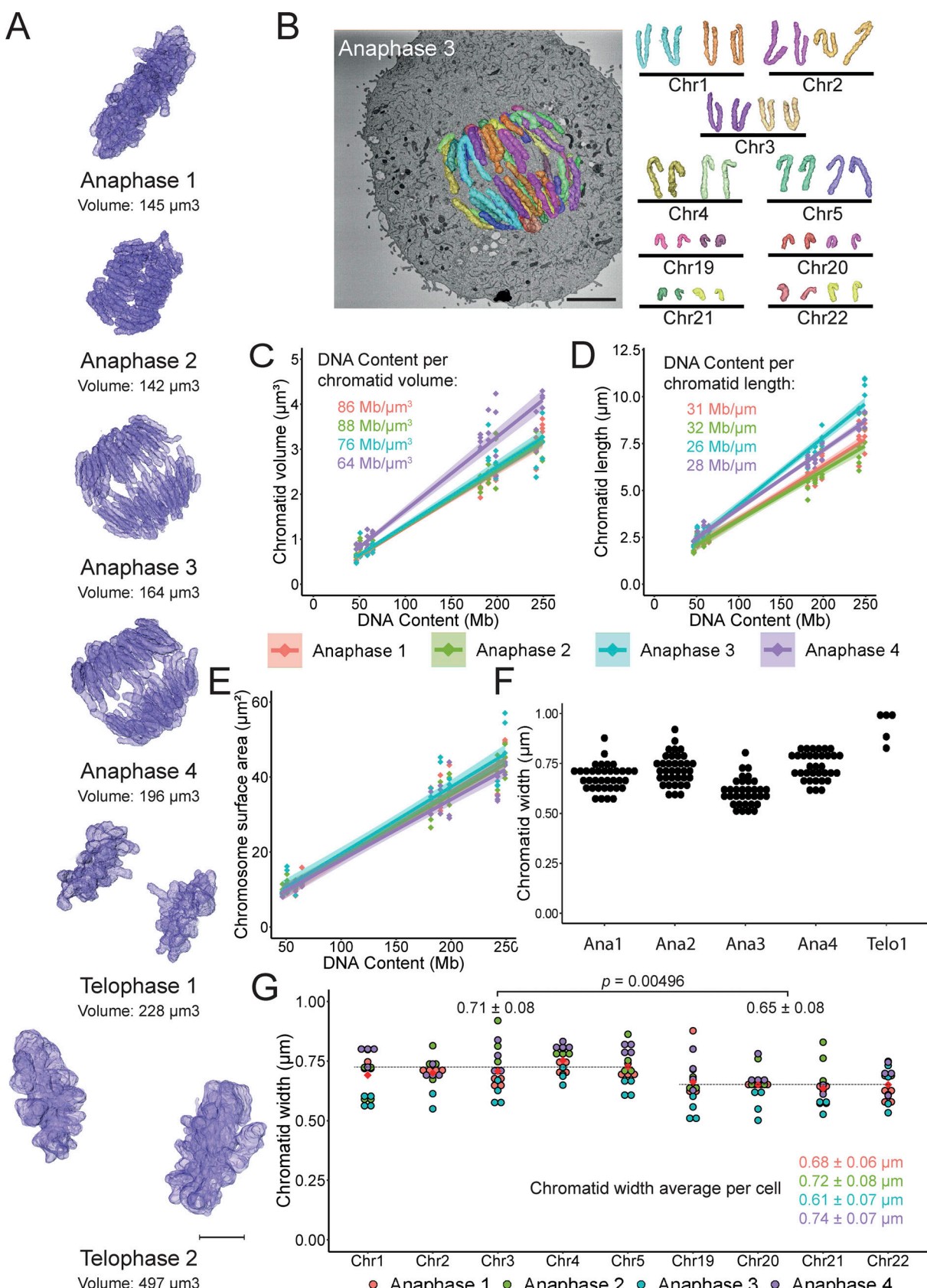

Figure 3. **Chromatid length and width vary as anaphase progresses. (A)** Models of four anaphase and two telophase cells with the total chromosome volume of each model annotated underneath each image. Scale bar = 5 µm. **(B)** Chromosome segmentations and orthoslice (left; scale bar = 5 µm). Separated sister chromatids are represented in the same color. Identification of chromatids 1–5 and chromatids 19–22 (right). **(C)** Correlation between the chromatid

volume and DNA content of the anaphase cells ($n$ = 4). **(D)** Correlation between the chromatid length and DNA content of the anaphase cells ($n$ = 4). **(E)** Correlation between the chromatid surface area and DNA content of the anaphase cells ($n$ = 4). **(F)** Chromatid width of anaphase cells and telophase 1 cell. Each dot represents the average of 10 measurements per chromatid. **(G)** Chromatid width is ordered by identified chromosomes (Chr1–5 and Chr 19–22). The average of 10 measurements per chromatid is plotted at each point. The average chromatid width and SD for all the chromatids is represented below the graph. The average width ± SD of large and small chromosomes are represented above the graph. Two-tailed Student's $t$ test, $n$ = 4 cells, P = 0.00496.

chromosome 1 remained similar to that in Anaphase 1, 1.01 ± 0.29 μm. This suggests that the chromosomes are stretching, possibly due to residual links (likely catenations) between telomeres. At this point, P2P increased from 14 to 17 μm, suggesting that this cell was in anaphase B with the increase in the sK2K distance driven by the pole separation.

A further increase in P2P distance to 21 μm occurred in Anaphases 3 and 4. This ~4 μm increase in P2P distance was accompanied by a ~2–3 μm increase in sK2K. Interestingly, comparing Anaphases 3 and 4, the sK2K distances were similar (10.1 ± 0.28 μm and 9.24 ± 1.23, respectively), but sT2T distances were quite different (0.72 ± 0.60 and 1.75 ± 0.92 μm, respectively) (Fig. 4 B). This suggested that there may be cell-to-cell variability in the resolution of telomeric sister chromatid interlinks (presumably catenations). Release of these links is apparently associated with a recoil of the chromatid which shortens and thickens.

After analyzing Chr1 in detail, we compared the overall behavior of the nine large and small (Chr1–5, 19–22) chromosomes for the different anaphase stages, correlating the distances between separating sister kinetochores and telomeres (sK2K distances and sT2T distances) on the large (Chr1–5; Fig. 4 D, large dots) and small chromosomes (Chr19–22; Fig. 4 D, small dots). Interestingly, we could already detect differences in the sT2T distances between the large and small chromosomes in the earliest Anaphase 1. In later anaphases, the kinetochores and the telomeres of the small chromosomes were consistently more separated than those of the large chromosomes. The larger telomere separation is expected, given that these chromosomes are shorter; however, the increased separation of the kinetochores indicates that the small chromosomes approach the spindle poles more closely.

Indeed, our analysis revealed that smaller chromosomes approach the spindle poles more rapidly and more closely than larger ones. We compared the sK2K distances versus the sK2P (kinetochore-to-pole) distances to see how chromosomes segregate to the spindle poles in the different anaphase cells. In early anaphase (Anaphase 1), large and small chromosomes showed similar sK2K and sK2P distances, with a <10% difference in distance. But, in later stages (Anaphases 2–4), large chromosomes showed shorter sK2K distances than small chromosomes (almost 17% shorter in Anaphase 4) (Fig. 4 E and Table S2). Thus, sister chromatids of larger chromosomes separated less than those of smaller chromosomes. Conversely, sK2P distances for large chromosomes were longer compared with those for small chromosomes (almost 18% longer in Anaphase 4) (Fig. 4 E and Table S2). Thus, small chromosomes move closer to the spindle poles and apparently do so more rapidly than large chromosomes.

Our observations suggest that small chromosomes may tend to be nearer to the poles because of their location in the metaphase plate. Typically, small chromosomes were situated toward the center of the metaphase plate, while large chromosomes were found toward the periphery (Booth et al., 2016; McIntosh and Landis, 1971; Mosgöller et al., 1991; Sajid et al., 2021; Takenouchi et al., 2024). During anaphase, small chromosomes maintain this central position and move closer to the spindle poles, while large chromosomes remain on the outer edges (see Fig. 4 F). This difference in chromosome positioning could help to give time and space for the large chromosomes to resolve catenations and segregate correctly. A recent study demonstrated that small chromosomes tend to move faster to the inner metaphasic plate. This could promote premature chromosome segregation due to bipolar microtubule forces, which are stronger in the inner region. As a consequence, the smaller chromosomes are more susceptible to weakened chromosome cohesion, resulting in segregation errors (Takenouchi et al., 2024).

## Near-millimolar concentration of nucleosomes in mitotic chromosomes

Remarkably, chromosomes establish a relatively constant volume (and therefore nucleosome density) by late prometaphase, which they maintain through early anaphase. This is consistent with Hi-C studies, which reveal that condensin II loops reach a maximum size of ~400 kb by the end of prophase and remain relatively constant after that (Samejima et al., 2024, *Preprint*).

In RPE1 hTERT CDK1as cells, the total volume of metaphase chromosomes is remarkably constant at 140 ± 4.8 μm³, yielding a DNA density of ~89 Mb/μm³. Assuming that each nucleosome has 195 bp of DNA (146 bp in two turns around the histone octamer plus 49 bp of linker DNA) (Tate and Philipson, 1979; Holde, 1989) (plus ATAC-seq results indicating that RPE1 cells likely only have 0.3% of the genome in nucleosome-free regions [Zou et al., 2024]), this yields an average packing density of 4.57 × 10⁵ nucleosomes per μm³. Thus, each nucleosome occupies a volume of 2,186 nm³ (13 × 13 × 13 nm).

To compare this value with previous results, we estimated the chromatin volume concentration (CVC), defined as the percentage of the total volume occupied by chromatin (Ou et al., 2017). In our studies, the calculated CVC in metaphase chromosomes is ~32%. This is an underestimate, as our calculation makes no allowance for the fact that nucleosomes are cylindrical solids linked by DNA carrying various bound proteins that occupy an unknown volume. Despite this caveat, our results are in remarkable agreement with calculations based on analysis of the local packing of nucleosomes stained by chromEMT—a CVC between 37% and 55% for heterochromatin domains and mitotic chromatin (Ou et al., 2017).

Interestingly, in a previously published study (Samejima et al., 2018) and in our own unpublished work (Samejima

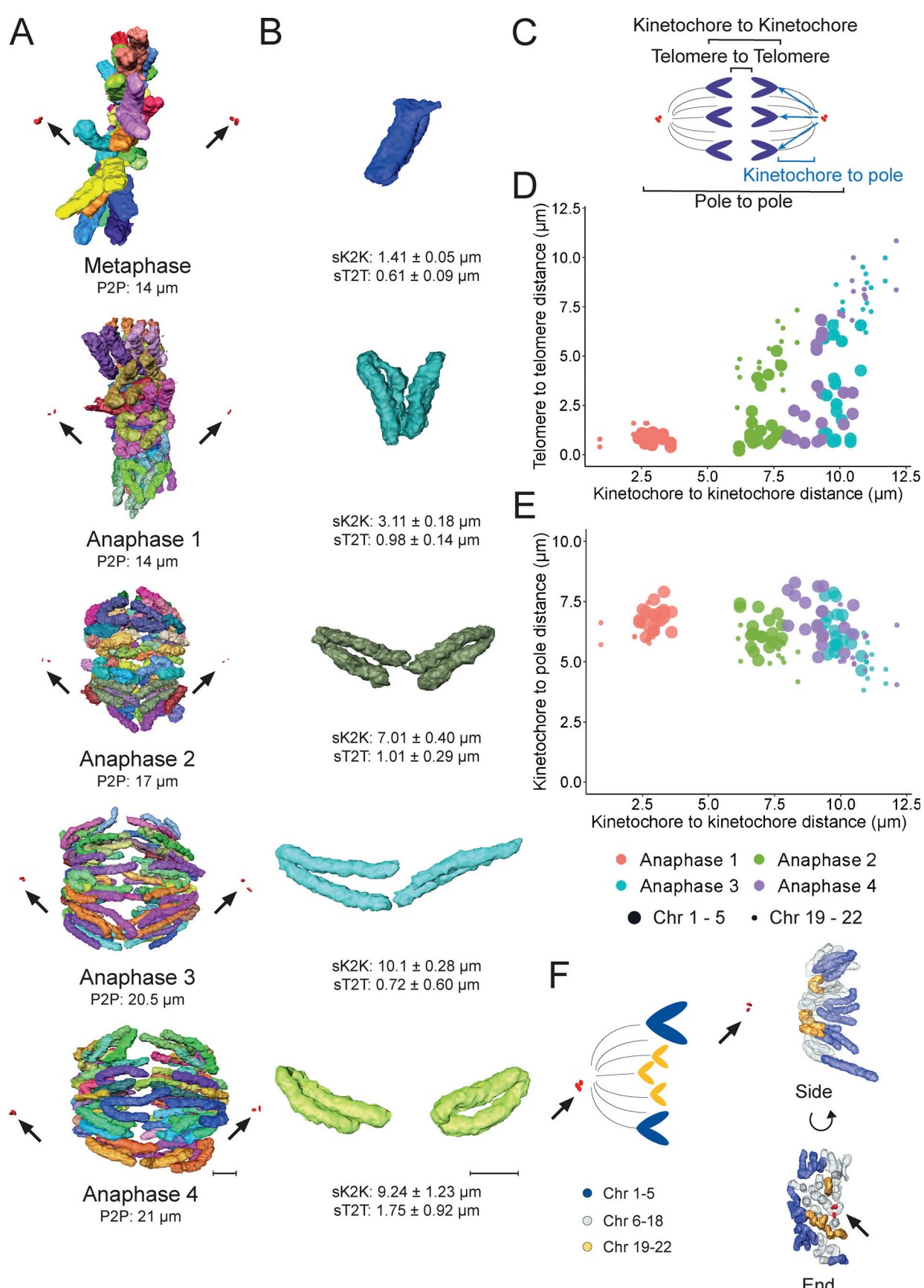

Figure 4. **Small chromatids usually occupy the spindle interior and approach the poles more closely than large chromatids. (A)** Representative images of separated chromosomes in one metaphase and four anaphase cells. Pole to pole (P2P) distance is shown under each image. Centrosomes are colored in red

(arrows). Scale bar = 2 µm. **(B)** Representation of chromosome 1 of each cell. Sister kinetochore to kinetochore distance (sK2K) and sister telomere to telomere distance (sT2T) are shown under each image. Scale bar = 2 µm. **(C)** Graphic representation of various distances measured: Kinetochore to kinetochore (K2K), telomere to telomere (T2T), kinetochore to pole (K2P), and pole to pole (P2P). **(D)** Correlation between K2K distance and T2T distance. **(E)** Correlation between K2K distance and K2P distance. Each individual cell is shown in a single color. Chromosomes 1–5 are represented by big dots while chromosomes 19–22 are represented by small dots. **(F)** Left: Graphic representation of the relative location of large and small chromosomes. Right: Anaphase 4 model showing large (blue) and small (yellow) chromosomes in side-view and end-on view. Centrosomes are represented in red, with arrows, in both panels.

et al., 2024, *Preprint*), we determined the volume of the chicken chromosome complement as 70–75 µm³ in metaphase DT40 cells. Interestingly, this yields a metaphase chromosome density of ~29 Mb/µm³, about one-third the density seen in RPE1 cells.

We were surprised to observe that overall structural parameters of mitotic chromosomes (total volume, DNA packing density in Mb/µm³, and chromosome and chromatid diameter) are remarkably constant from late prometaphase through early anaphase (Fig. 5). Therefore, mitotic chromosome formation appears to culminate in a relatively defined (plateau) density of nucleosome packing. We and others had generally assumed that chromatin packing density varies across the cell cycle peaking in anaphase and decreasing thereafter (Pederson, 1972; Ma et al., 2019). It was proposed that this cycle is driven at least in part by the assembly or enzymatic functions of a non-histone scaffold (Adolph et al., 1977b; Laemmli et al., 1978; Paulson et al., 2021). Although scaffold proteins clearly have a role in shaping mitotic chromosomes (Ono et al., 2003; Hudson et al., 2003; Gibcus et al., 2018; Poonperm et al., 2017; Yoshida et al., 2022), it could be argued that the density plateau observed in our study is more consistent with mitotic condensation being achieved by a polymer melt (Nishino et al., 2012) or phase separation mechanism (Gibson et al., 2019, 2023; Schneider et al., 2022), possibly linked to changes in histone posttranslational modifications (Zhiteneva et al., 2017; Schneider et al., 2022).

Is it reasonable that mitotic chromosome condensation could be driven by a form of phase separation? In one recent study, nucleosome 12-mers were found to undergo phase separation in vitro in the presence of cations, yielding droplets with a nucleosome concentration of ~340 µM (Gibson et al., 2019). Remarkably, the average concentration of nucleosomes estimated here in human metaphase chromosomes is more than twice this $(4.57 \times 10^5$ nucleosomes/µm³) × (1 × 10¹⁵ µm³/liter) × (1 × 10⁶ µM/M)/(6.02 × 10²³ nucleosomes/M) = 760 µM.

In the in vitro study, histone hyperacetylation was found to dissolve the chromatin droplets, and therefore the higher concentration of nucleosomes seen in metaphase chromosomes could be due to a different modification status of the histones (Zhiteneva et al., 2017; Schneider et al., 2022). Indeed, differing levels of residual acetylation could possibly explain why chicken mitotic chromosomes have a roughly threefold lower concentration of nucleosomes than their human counterparts do.

**Implications**

We report here that human chromosomes reach an apparent plateau concentration of nucleosomes between late prometaphase and early anaphase—i.e., while chromosome segregation is ongoing. The fact that this concentration appears to be a plateau, rather than a peak, invites the speculation that the final nucleosome concentration approaches a limit—perhaps due to

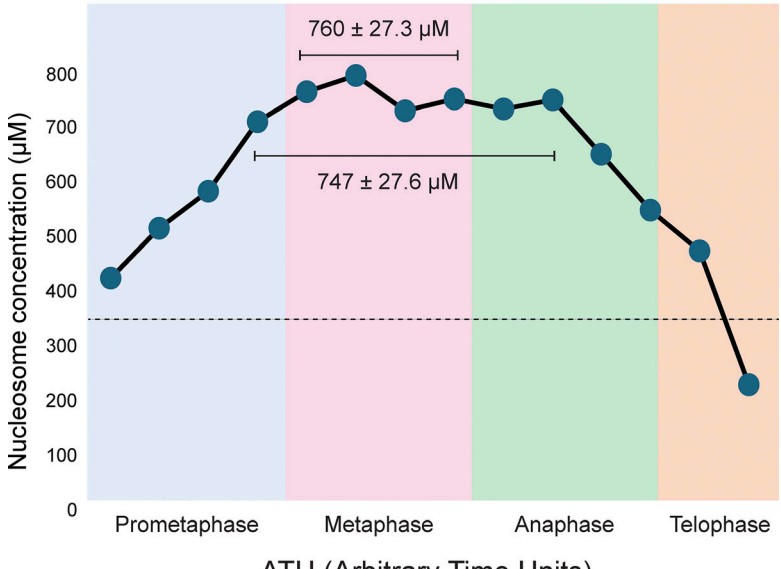

Figure 5. **Nucleosome density in mitotic chromosomes reaches a plateau in the near-millimolar range that remains relatively constant from late prometaphase through early anaphase.** This plateau density is well above that calculated for nucleosome droplets in an in vitro chromatin phase (340 µM, dotted line; Gibson et al., 2019). Each circle represents the nucleosome concentration in a cell in which the entire chromosome volume has been measured in a 3View tomogram. The temporal order of the cells in prometaphase and anaphase/telophase was established by measuring the centrosome separation. The temporal order of the four metaphase cells is not known. Y axis, nucleosome concentration (µM). X axis, ATU (arbitrary time units).

phase separation (Gibson et al., 2019, 2023). This has implications for current studies of loop extrusion by various SMC motors. The condensin motor is reported to be very fast but also very weak (Ganji et al., 2018; Prevo and Earnshaw, 2024). Whether it can exert sufficient force to function in a highly concentrated putative nucleosome phase as it does in a dilute solution remains an interesting and challenging question for future studies.

## Materials and methods

### Cell culture and generation of stable cell lines
RPE1 hTERT (Cat Number: CRL-4000; ATCC) cells were grown in DMEM media (Life Technologies) supplemented with 10% SFB at 37°C with 5% $CO_2$. To obtain the RPE1 CDK1[as] cell line, we electroporated $5 \times 10^5$ RPE1 hTERT cells with hCas9, gRNA (CDK1), and Sleeping Beauty with *X. laevis* CDK1[as] gene plasmids using a Neon system (Life Technologies) according to the manufacturer's protocol.

All cell lines were authenticated using STR profiling and were tested negative for mycoplasma using a LookOut mycoplasma PCR Detection Kit (Sigma-Aldrich) following the manufacturer's protocol.

To obtain cells in mitosis, we presynchronized the RPE1 CDK1[as] cells by incubating them for 24 h with 200 nM palbociclib (SelleckChem). 12 h after palbociclib wash-out, cells were incubated for 8 h with 0.5 μM 1NM-PP1 (SelleckChem) to block them in late $G_2$. Finally, wash-out of the 1NM-PP1 and incubation of cells with conditioned media allowed them to enter mitosis.

### Immunoblotting
Cells were collected and lysed using Lysis Buffer (#9803; Cell Signaling) supplemented with a protease inhibitor cocktail (#5871; Cell Signaling). Lysates were sonicated and centrifuged for 15 min at 20,000 × *g* at 4°C. The supernatants were collected and supplied with Laemmli sample buffer and boiled at 96°C for 5 min. 20 μg of protein extract was loaded in a 10% SDS-bisacrylamide gel and transferred to a PVDF membrane. The membrane was blocked with milk and then incubated with antibodies recognizing CDK1 (ab18; Abcam) and α-tubulin (05-829; Merck). After washing out the primary antibodies, the membrane was incubated with a secondary antibody conjugated to Alexa Fluor Plus 680 and 800 (A32802, A32730; Invitrogen), imaged using Licor Odyssey CLx, and analyzed with ImageStudio Software (Licor).

### Flow cytometry
A total of $3 \times 10^4$ cells per cm² were seeded on six-well plates and synchronized in the $G_2$ phase as described above. After palbociclib treatment, the cells were washed in PBS, trypsinized, fixed in 70% cold ethanol, and incubated at –20°C for at least 12 h. The cells were then washed once, resuspended in PBS, and stained with 10 μg/ml propidium iodide in 1.1% sodium citrate buffer supplemented with 0.25 mg/ml RNase A for 1 h in the dark. Data were collected with a FACSCalibur flow cytometer (Becton Dickinson), and the distribution of DNA content was analyzed with CellQuest Pro Software (Becton Dickinson). Cell cycle phases were determined using ModFit LT software (Becton Dickinson).

### Mitotic index
RPE1 CDK1[as] cells were seeded on coverslips at 60% confluence. After the synchronization protocol, as described above, the cells were fixed with 4% formaldehyde in PBS for 8 min. They were then washed, stained with Hoechst, and mounted on a slide. We evaluated the chromosome morphology and scored at least 1,000 cells per condition.

### Chromosome spreads and karyotype
RPE1 hTERT and RPE1 CDK1[as] were treated with 0.2 mg/ml colcemid (KaryoMAX; Invitrogen) for 1 h before harvesting. After trypsinization, the cells were treated with hypotonic solution (0.8% Sodium citrate) for 30 min at 37°C and fixed with Carnoy's solution (Methanol/Acetic acid 3:1). To count the number of chromosomes, the cells were spread on coverslips and stained with DAPI. 50 cells were counted per coverslip.

For karyotype analysis, we spread the cells on coverslips, stained G banding using Giemsa and Wright solutions, and mounted the cells using Entellan mounting medium. The G-band pattern of 20 cells was analyzed for each cell clone.

### Electron microscopy sample preparation
RPE1 CDK1[as] cells were seeded on a MatTeck dish with coordinates and synchronized according to the established protocol. We fixed the cells with 2% glutaraldehyde and 2% formaldehyde in 0.1 M sodium cacodylate buffer for 1 h (pH 7.4). We stained the DNA with Hoescht in 0.1 M cacodylate buffer for 30 min.

To identify cells of interest, we placed the fixed cells on a wide-field DeltaVision Elite (Applied Precision) microscope with PCO edge 4.2 sCMOS camera and 100× NA 1.45 UPLX Apochromat objective with oil immersion (refractive index = 1.514) and Cy5 filter set (640 ± 30) using the SoftWoRx 3.6 (Applied Precision) software. We also acquired images using the 20× NA 0.5 UPlanFl objective dry to identify the ROI and record the coordinates on the MatTeck coverslip.

We next rinsed the cells 5 × 2 min with 0.1 M sodium cacodylate buffer and stained them for 1 h with 2% osmium tetroxide and 1.5% potassium ferrocyanide. Next, we added 1% tannic acid as a mordant for 20 min at room temperature, followed by a second osmification, adding 2% osmium tetroxide for 1 h. The samples were then incubated overnight with 1% uranyl acetate in aqueous solution. Finally, we added Walton's lead aspartate (0.02 M lead nitrate in 0.03 M L-aspartic acid, adjusted to pH 5.5) for 30 min at 60°C. We rinsed the cells 5 × 2 min with $ddH_2O$ between each step. We next dehydrated the sample in a graded ethanol series of 30%, 50%, 70%, and 90% in $ddH_2O$ for 5 min each, followed by 2 × 5 min 100% ethanol. Samples were then infiltrated with Agar 100 Hard Premix resin (Agar Scientific) at a 1:1 ratio (resin: 100% ethanol) and then, resin-only for 30 min. We then embedded the samples in 1.5 mm of 100% fresh resin and cured them for 48 h at 60°C.

To prepare the blocks, we removed the resin from the MatTek dish using pliers. We identified the cells of interest using a stereoscopic microscope and the coordinates on the coverslip. We then

cut the block with a junior hacksaw and glued it to a cryopin (cell side up) using silver epoxy resin. Then, the block was trimmed using a microtome to select the region of interest. The samples were coated with 10 nm gold/platinum before imaging.

### SBF–SEM imaging and data acquisition
Images were acquired using a Gatan 3View serial block-face system (Gatan) installed on a FEI Quanta 250 FEG scanning electron microscope (FEI Company). Images were collected with the following parameters: a pixel size depending on the cell size, magnification 5.7K, and a chamber pressure of 70 Pa at 4 kV.

### 3D reconstruction and image analysis
The images for 3D reconstruction were binned in a 5 × 5 × 1 (x,y,z) ratio using the average method in ImageJ. They were then transformed into 8-bit images. Further preprocessing, including aligning the sections, was done in AMIRA software. We used the difference between the cytoplasm and chromosome grey values to select the threshold to identify chromosomes (Fig. 2 A). The individual chromosomes were separated using the separate objects module using 3D interpretation, 26 neighborhood, and 4-marker extension.

All EM data are available from EMPIAR—Accession code: EMPIAR-11953.

### Statistical analysis
For chromosome/chromatid width analysis, we averaged 10 measurements per chromosome per cell. We assessed the normality of the data using the Shapiro-Wilk test. Our data showed a P value >0.05, indicating a normal distribution. Then, we conducted a two-tailed Student's $t$ test to determine any significant differences.

Linear regressions were performed using R and analyzed using ANOVA test. Data distribution was assumed to be normal, but this was not formally tested.

### Analysis of ATAC-seq data for RPE1 cells
An estimate of the proportion of RPE-1 DNA contained within nucleosome-free regions was calculated from ATAC-seq "peak" data obtained from ChIP-Atlas (Zou et al., 2024) (track type class; ATAC-seq, cell type class; neuronal, threshold for significance; 200, cell type; hTERT RPE-1). Overlapping peaks were merged using "bedtools merge," and the proportion of reads in "peaks" (4.7 Mbp) were compared with the total length of non-repetitive DNA in the hg38 genome build (1.52 Gbp), giving a value of 0.3% of the genome being localized to nucleosome depleted regions.

### Online supplemental material
Fig. S1 shows the establishment of the CDK1[as] synchronization system in RPE1 hTERT cells and the characterization of the clones (Characterized the RPE1 CDK1as cell line and described synchronization in mitosis). Fig. S2 shows that the excess blocking time in G[2] increases the number of multipolar cells (Explained the high number of apoptotic cells after a long block with 1NM-PP1). Fig. S3 shows that the resolution of the image has little effect on the volume but a substantial effect on the surface area of the chromosomes. Fig. S4 shows the correlation between chromosome/chromatid width and chromosome/chromatid length in metaphase and anaphase cells. Table S1 shows chromosome measurements in different mitotic stages (Related to Figs. 1, 2, and 3). Table S2 shows the difference between large and small chromosome distances from the spindle pole in anaphase (Related to Fig. 4). Table S3 shows nucleosome volume calculation (Related to Fig. 5).

### Data availability
Cell lines are available from the corresponding author, William C Earnshaw, upon request. EM data has been deposited in EMPIAR (accession code: EMPIAR-11953). Constructs in this study are available from Addgene (ID: 118597 and 118596).

## Acknowledgments
We thank Kumiko Samejima for her advice in establishing the CDK1[as] system and Adriana Gudiño for her advice on the chromosome spreads and karyotype protocols.

This work was funded by Wellcome grants 107022 and 221044 to W.C. Earnshaw and 203149 to the Wellcome Centre for Cell Biology. I.A. Prior is supported by a North West Cancer Research endowment (CR1166). D.G. Booth is funded by a Biotechnology and Biological Sciences Research Council (BBSRC) David Phillips Fellowship (BB/V005626/1), Royal Society Research Grant (RGS/R2/202366), and Leverhulme Trust Project Grant (RPG-2021-118). E. Simpson is funded by a University of Nottingham BBSRC PhD studentship. Open Access funding provided by University of Edinburgh.

Author contributions: F. Cisneros Soberanis: Conceptualization, Formal analysis, Investigation, Methodology, Supervision, Visualization, Writing—original draft, Writing—review & editing, E.L. Simpson: Data curation, Formal analysis, Investigation, Methodology, Visualization, Writing—review & editing, A.J. Beckett: Data curation, Methodology, Resources, Writing—review & editing, N. Pucekova: Formal analysis, S. Corless: Formal analysis, Validation, N.Y. Kochanova: Formal analysis, Writing—review & editing, I.A. Prior: Investigation, Supervision, Writing—review & editing, D.G. Booth: Conceptualization, Funding acquisition, Methodology, Resources, Supervision, Validation, Writing—review & editing, W.C. Earnshaw: Conceptualization, Funding acquisition, Resources, Supervision, Writing—original draft, Writing—review & editing.

Disclosures: The authors declare no competing interests exist.

Submitted: 27 March 2024

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

# Supplemental material

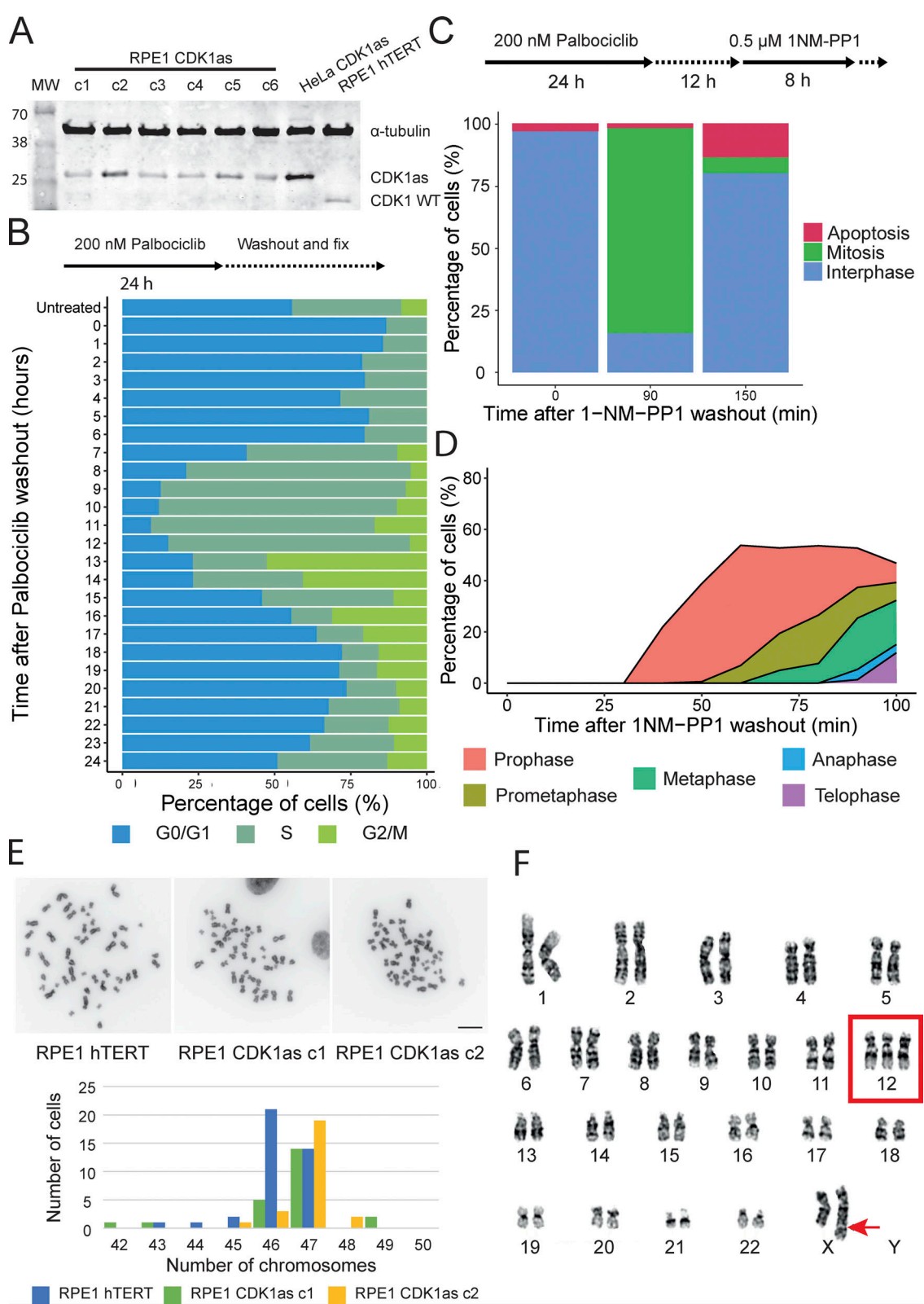

Figure S1. **Establishing the CDK1<sup>as</sup> synchronization system in RPE1 hTERT cells and characterization of the clones. (A)** Western blot to detect CDK1as protein in the RPE1 CDK1<sup>as</sup> clones, parental RPE1 hTERT (left), and HeLa CDK1<sup>as</sup> (right) are shown as control. **(B)** Palbociclib block and release treatment for 24 h showing cells in G0/G1 (blue), S phase (dark green), and G2/M (light green). **(C)** Synchronization protocol (top) with the percentage of cells in interphase (blue), mitosis (green), and apoptosis (red) at 0, 90, and 150 min after the 1NM-PP1 washout. **(D)** Mitotic index counting the relative number of various mitotic stages. **(E)** Counts of the number of chromosomes in the RPE1 hTERT (parental cell line) and the 2 RPE1 CDK1<sup>as</sup> clones, n = 20–50. Scale bar = 10 µm. **(F)** Representative karyotype of RPE1 CDK1<sup>as</sup> clone 2 showing an extra chromosome 12 and an insertion in chromosome X, n = 20. Source data are available for this figure: SourceData FS1.

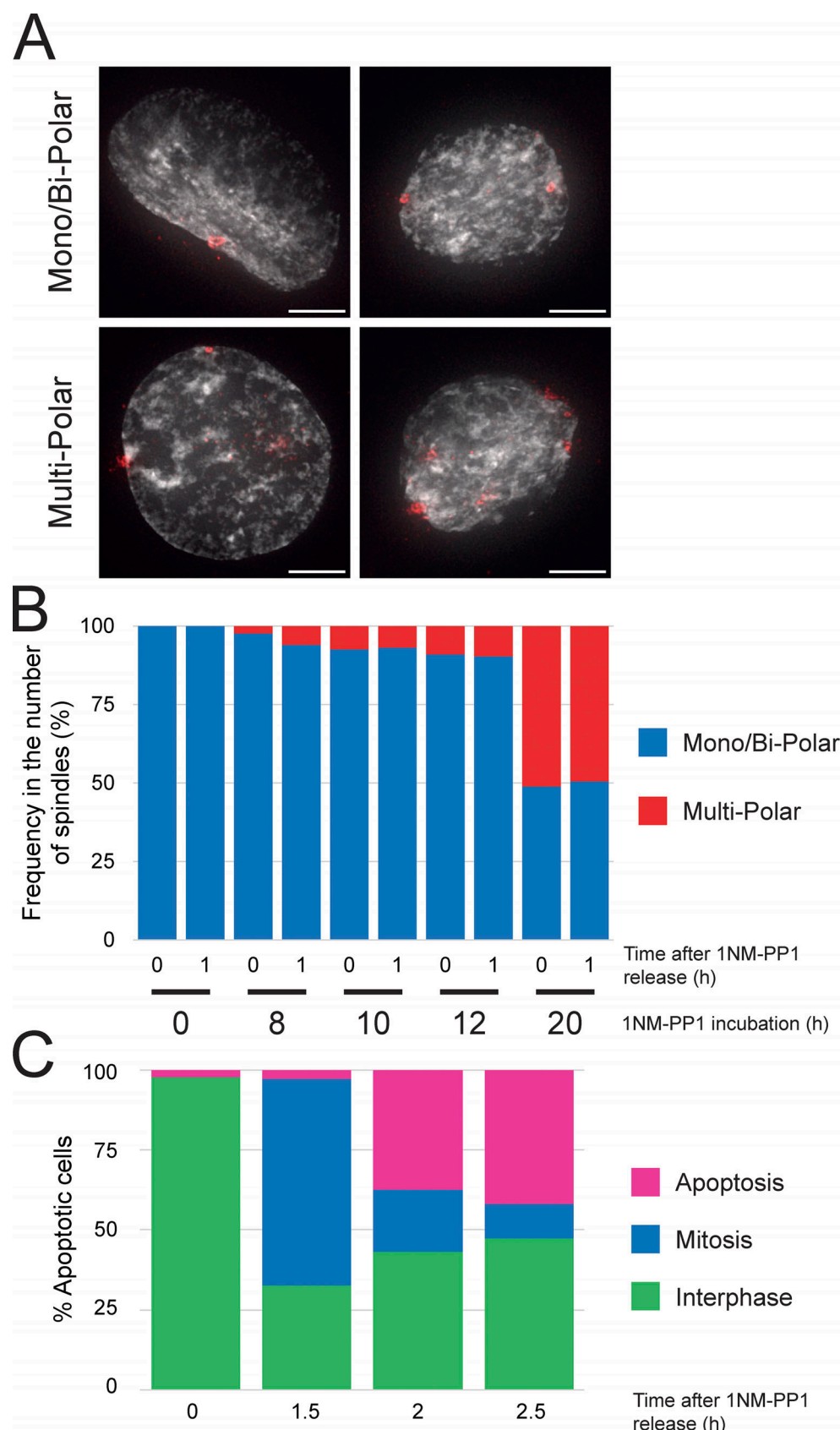

Figure S2. **Excess blocking time in G$_2$ increases the number of multipolar cells. (A)** Representative images of monopolar, bipolar, and multipolar RPE1 CDK1$^{as}$ cells. Scale bar = 5 µm. **(B)** Frequency of number of spindle poles (mono/bipolar and multipolar) after 8, 10, 12, and 20 h incubation with 1NM-PP1. **(C)** Percentage of apoptotic cells (0, 1.5, 2, and 2.5 h) after release from a 20 h incubation with 1NM-PP1. Each represents the time after release.

High Resolution ◄————————————————————► Low Resolution

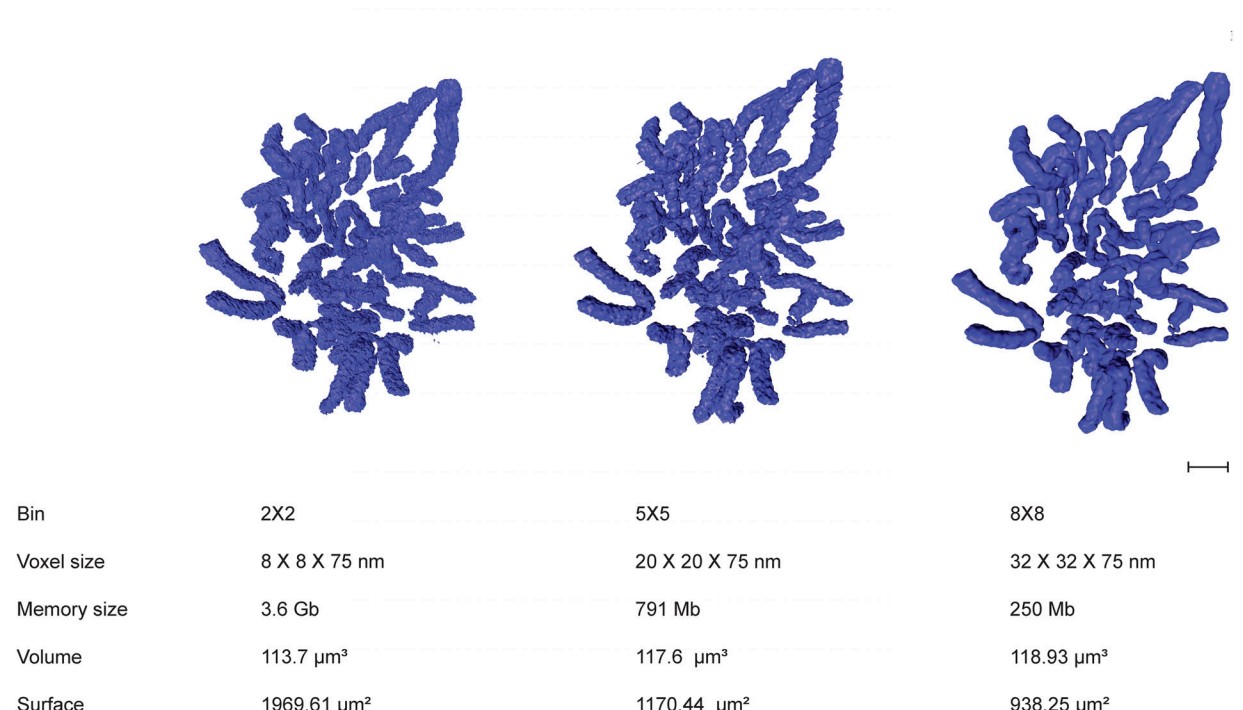

| Bin | 2X2 | 5X5 | 8X8 |
|---|---|---|---|
| Voxel size | 8 X 8 X 75 nm | 20 X 20 X 75 nm | 32 X 32 X 75 nm |
| Memory size | 3.6 Gb | 791 Mb | 250 Mb |
| Volume | 113.7 µm³ | 117.6 µm³ | 118.93 µm³ |
| Surface | 1969.61 µm² | 1170.44 µm² | 938.25 µm² |

Figure S3. **The resolution of the image has little effect on the volume but a substantial effect on the surface area of the chromosomes.** High-resolution versus low-resolution models of the same cell are represented at the top. Scale bar = 2 µm. Voxel size, memory size, chromosome volume, and chromosome surface area information are represented below each image.

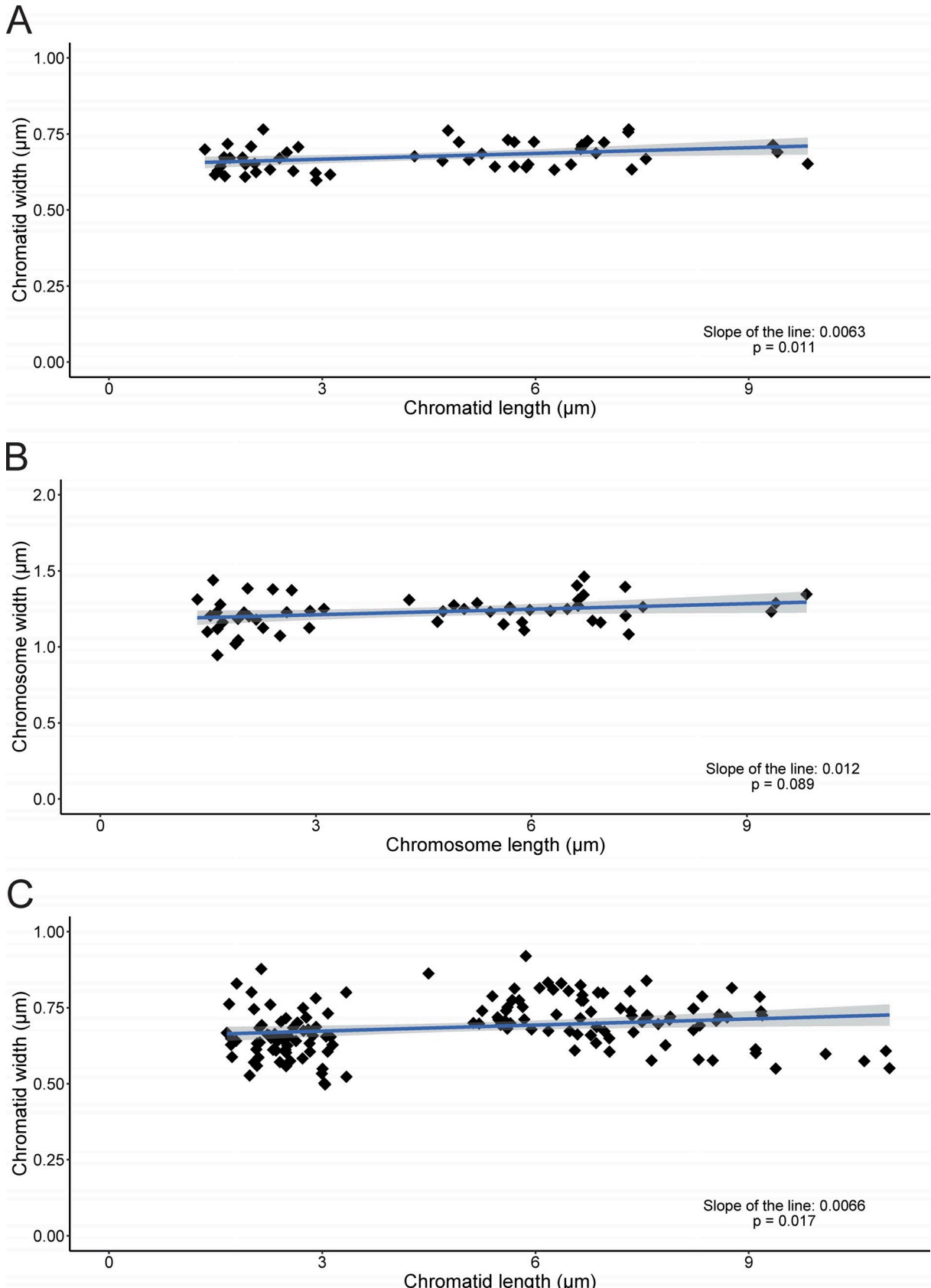

Figure S4. **Correlation between chromosome/chromatid width and chromosome/chromatid length in metaphase and anaphase cells. (A–C)** Metaphase chromatid width, (B) Metaphase chromosome width, and (C) Anaphase chromatid width. Each point represents the average of 10 measurements per chromosome. The slope of the line (blue) and P values obtained from an ANOVA test are annotated in each graph.

Provided online are Table S1, Table S2, and Table S3. Table S1 shows chromosome measurements in different mitotic stages. Table S2 shows the difference between large and small chromosome distances from the spindle pole in anaphase. Table S3 shows nucleosome volume calculation.

