## [Peer Review File · The Journal of Cell Biology]

Near Millimolar Concentration of Nucleosomes in Mitotic Chromosomes from Late Prometaphase

Fernanda Cisneros Soberanis, Eva Simpson, Alison Beckett, Nina Pucekova, Samuel Corless, Natalia Kochanova, Ian Prior, Daniel Booth, and William Earnshaw

Corresponding Author(s): William Earnshaw, University of Edinburgh and Fernanda Cisneros Soberanis, University of Edinburgh

Review Timeline:

Submission Date:	2024-03-27
Editorial Decision:	2024-06-20
Revision Received:	2024-07-05
Editorial Decision:	2024-07-23
Revision Received:	2024-07-29

Monitoring Editor: Hironori Funabiki

Scientific Editor: Dan Simon

Transaction Report:

DOI: <https://doi.org/10.1083/jcb.202403165>

June 20, 2024

Re: JCB manuscript #202403165

Prof. William C Earnshaw
University of Edinburgh
Institute of Cell Biology
University of Edinburgh, Swann Building, Kings Buildings
Max Born Crescent
Edinburgh EH9 3BF
United Kingdom

Dear Prof. Earnshaw,

Thank you very much for submitting your manuscript, "Near Millimolar Concentration of Nucleosomes in Human Mitotic Chromosomes from Late Prometaphase into Anaphase" to JCB. The manuscript was assessed by three reviewers, whose comments are appended to this letter.

As you can see, the opinions of the reviewers are split. While Reviewers 1 and 3 question the novelty of your manuscript, Reviewer 2 recognizes the importance of your efforts to systematically measure the mitotic chromosome volume during mitotic progression using SBF-SEM. As my assessment is more in line with the opinion with Reviewer 2, I would like to invite submission of a revised manuscript. However, the reviewers pointed out several technical issues that I feel important to be clarified. Particularly, along with a list of minor points listed with Reviewer 2, please respond to points #2 and #3 raised by Reviewer 1, questioning if you could make the conclusion by the low sample numbers. I hope that it will be manageable to you and your coauthors.

GENERAL GUIDELINES:

Text limits: Character count for a Report is < 20,000, not including spaces. Count includes title page, abstract, introduction, the joint Results & Discussion, and acknowledgments. Count does not include materials and methods, figure legends, references, tables, or supplemental legends.

Figures: Reports may have up to 5 main text figures. To avoid delays in production, figures must be prepared according to the policies outlined in our Instructions to Authors, under Data Presentation, <https://jcb.rupress.org/site/misc/ifora.xhtml>. All figures in accepted manuscripts will be screened prior to publication.

*****IMPORTANT:** It is JCB policy that if requested, original data images must be made available. Failure to provide original images upon request will result in unavoidable delays in publication. Please ensure that you have access to all original microscopy and blot data images before submitting your revision. ***

Supplemental information: There are strict limits on the allowable amount of supplemental data. Reports may have up to 3 supplemental figures. Up to 10 supplemental videos or flash animations are allowed. A summary of all supplemental material should appear at the end of the Materials and methods section.

Please note that JCB now requires authors to submit Source Data used to generate figures containing gels and Western blots with all revised manuscripts. This Source Data consists of fully uncropped and unprocessed images for each gel/blot displayed in the main and supplemental figures. If your revised manuscript will include cropped gel and/or blot images, please be sure to provide one Source Data file for each figure that contains gels and/or blots along with your revised manuscript files. File names for Source Data figures should be alphanumeric without any spaces or special characters (i.e., SourceDataF#, where F# refers to the associated main figure number or SourceDataFS# for those associated with Supplementary figures). The lanes of the gels/blots should be labeled as they are in the associated figure, the place where cropping was applied should be marked (with a box), and molecular weight/size standards should be labeled wherever possible. Source Data files will be made available to reviewers during evaluation of revised manuscripts and, if your paper is eventually published in JCB, the files will be directly linked to specific figures in the published article.

The typical timeframe for revisions is three to four months. If you anticipate any difficulties in meeting this aforementioned revision time limit, please contact us and we can work with you to find an appropriate time frame for resubmission. Please note that papers are generally considered through only one revision cycle, so any revised manuscript will likely be either accepted or rejected.

Thank you for this interesting contribution to Journal of Cell Biology. You can contact us at the journal office with any questions at cellbio@rockefeller.edu.

Sincerely,

Hironori Funabiki, PhD
Monitoring Editor
Journal of Cell Biology

Dan Simon, PhD
Scientific Editor
Journal of Cell Biology

Reviewer #1 (Comments to the Authors (Required)):

In this manuscript, authors have used quantitative 3D electron microscopy to visualize and characterize changes to chromosome volume, surface area, width and length during mitotic progression between prometaphase through telophase in human RPE1 cells. The main conclusion is that chromosome volume reaches a plateau by prometaphase and chromosome volume remains constant during metaphase. Using the measured volume, they estimate a nucleosome concentration of ~666 micromolar, which is above the concentration for nucleosomes to undergo phase separation as determined in previous studies.

This is a nice, descriptive study of chromosome volume changes during mitosis. The manuscript is well written and the results are clearly presented. I enjoyed reading.

My major comments are as follows:

1. The manuscript is very descriptive and does not offer new mechanistic insight into chromosome compaction/condensation during mitosis. It is not clear what are the mechanisms required to maintain the constant volume measured during metaphase or why this is functionally important.
2. Some of the conclusions rely on the accuracy of volume and surface area measurements, however, as the authors demonstrate, these measurements (particularly surface area/granularity) are heavily influenced by the pixel size and resolution they pick.
3. Only a handful of chromosomes are measured in each phase of mitosis and all the major conclusions rely on these statistically low and underpowered datasets.
4. Major conclusions are speculative and depend on the accuracy of measurements (see comments 2 and 3). It is not clear what is the margin of error on the nucleosome concentration estimates, which is the main conclusion of the manuscript and the title of the manuscript. I think the title is not fully supported by the speculative, back of the envelope calculations based on a few chromosomes.

Reviewer #2 (Comments to the Authors (Required)):

The manuscript by Fernanda Cisneros-Soberanis et al. presents a quantitative analysis of chromatin concentration measured during the progression through mitosis in RPE1 cells of human origin. In this study, the authors have further optimized their cell synchronization system, previously introduced in Gibcus et al. (2018), by implementing a presynchronization step. This step reduced the frequency of apoptosis and enabled the efficient synchronization of cells at specific mitotic stages, a feat not previously achieved to the best of my knowledge.

Additionally, correlative light and electron microscopy (CLEM) was employed to accurately determine the stage of mitosis for each cell. Subsequently, high-resolution data were acquired through serial block face scanning electron microscopy (SBF-SEM). Leveraging this advanced experimental setup, the authors investigated fundamental questions regarding chromatin compaction and decompaction during mitosis.

Their analysis unveiled an intriguing phenomenon: the density of mitotic chromatin remains remarkably constant and high from late prometaphase to early anaphase, which the authors term the "density plateau". This observation contradicts previous findings from light microscopy methods and offers essential novel insights into the organization and dynamics of mitotic chromatin.

SBF-SEM allows high-resolution data acquisition, but its throughput is much lower. Therefore, the considerable effort expended to record and reconstruct appreciable numbers of complete mitotic cells in 3D with impressive resolution, as shown in this study, should be appreciated.

The authors openly acknowledge the potential artifacts associated with the methods utilized, highlighting the necessity of cryo-electron volumetric imaging for overcoming these limitations. However, this technique is currently in its early developmental stage and cannot be applied. In their previous works, authors show that no discernable reduction of chromosome volumes occurs during sample preparation used in this work. Importantly, the key observation of the authors is based on a comparative analysis of different mitotic stages, which a possible minor structure cannot affect.

The study combines a carefully organized series of experiments, resulting in valuable data. Results are described and supplemented with clear overview images and graphs. In summary the presented work is original and significant.

Suggested minor improvements:

1. Lines 74-76. "They concluded that the maximal chromosome compaction occurs in late anaphase by an axial shortening of the chromosome arms. (Note, however, that the volume of a cylinder varies linearly with respect to length, but with the square of the radius.)" The reader would appreciate an explanation of this point. Does it mean that the radius measurements were not provided by Mora-Bermudez et al., therefore their statement is not conclusive?
2. Lines 186: "This technique allows us to achieve a resolution in X and Y of 4 nm... " Do you mean sampling step? The resolution of SEM is not directly linked to the pixel size, but depends of the electron optics quality, voltage, and likely the detector as you mention later.
3. Lines 238-240 "This ~20% difference in volume is likely due to the combination of improved resolution of a new detector in the 3View system..." This statement can be supported by providing resolution measurements for the previous and current detector performed on any standard sample. This can also be illustrated by showing the difference in density profile between the previous and current data (as shown in Figure 2A). The lower resolution will be evident in the smoother transition of gray values between the cytoplasm and chromosome.
4. Lines 278-283. The observation of the evolution of the shape of chromosome or chromatid profiles during mitosis is very interesting. Having a figure illustrating this interpretation and discussion would be useful for a better understanding, especially because the current study investigates more phases of mitosis than the previous one. Could it be compatible with and/or support the compaction hypothesis based on phase separation?
5. Do you see only surface granularity in prometaphase chromosomes, or are there also internal structural irregularities - cavities that have been demonstrated in some cell lines? (for example in Schroeder-Reiter E., et al. J Struct Biol. 2009;165:97-106).
6. Lines 340-354. You talk about the characteristics of the largest chromosome in prometaphase, and then move on to chromosome 1 in the context of the prometaphase-metaphase transition (Line 351). This sounds confusing. I understood that unambiguous chromosome recognition was not possible in prometaphase? Could you please clarify this?
7. Lines 355-358. "Thus, the linear density of chromatin appears to remain relatively constant as chromatids shorten during early mitosis." As I understand it, linear density is determined solely by chromatid length (Mb/ μm). How can it remain constant if chromatids become shorter, or am I misunderstanding?
8. Samejima et al, 2024, missing citation.
9. Statistical analysis. Using non-parametric tests like the Kolmogorov-Smirnov test would be more suitable for comparing relatively small distributions, especially when their normality cannot be justified.

Reviewer #3 (Comments to the Authors (Required)):

Summary: Cisneros-Soberanis et al. present a detailed quantitative analysis of chromosome compaction, which is essential for accurate chromosome segregation during mitosis, using Volume Electron Microscopy. Their research maps chromosomes with nanometer precision from prometaphase to telophase in human RPE1 cells. Notably, during prometaphase, the chromosomes exhibit primary constriction, leading to a smoother surface and shortened arms. The study highlights significant progressive compaction by late prometaphase that remains consistent through metaphase and early anaphase. Additionally, it is observed that chromosome volume increases just before the nuclear envelope reformation in late anaphase. These results support the proposed chromatin condensation in mitosis reaches and maintains a constant density threshold from late prometaphase through early anaphase.

Contribution to Mitotic Chromosome Formation Studies: This study supports previous theories proposed by Cremer (2003), Sajid (2021), and Beel (2021) concerning the importance of size and density in the formation and positioning of mitotic chromosomes. Although the study does not introduce novel information, it provides an intriguing speculation that the observed density limits during chromosome compaction could be related to phase separation. This hypothesis gains importance considering its innovative potential in the field, which has been further explored in recent studies by Gibson (2023 and 2019).

Overall, the study does not add novelty to the field at this publication stage. However, the field is open to many questions, encouraging the authors to pursue these lines of inquiry further in the study.

Potential for Broader Impact through Comparative Analysis: This study could significantly benefit from extending its analysis to compare stable karyotypical mammalian endothelial RPE1 cells with cancer cells. This comparison aims to identify variations in chromosome structure and the stability of nucleosome concentrations during mitosis. The use of RPE1 hTERT CDK1as cells as the model is well-justified, providing an advantage over alternatives like nocodazole. However, introducing motor protein inhibitors such as CENP-E or even exploring the role of SMC motors in DNA extrusion could offer novel insights into the forces and error correction mechanisms that influence chromosome structure during mitotic compaction.

Further Research Considerations: Including a control group of mitotic cells that are not synchronised could shed light on whether the slower progression into mitosis of the RPE1 hTERT CDK1as cells impacts chromosome compaction.

Methodological Enhancements: It would be beneficial to display the flow cytometry DNA content distribution within the cell cycle in Figure 1C and provide clear examples to delineate each phase in Figure 1D. Additionally, the authors should present the nucleosome imagery used for quantification alongside a 3D tomographic visualisation of each mitotic phase to enhance the understanding and reproducibility of their findings.

**WELLCOME TRUST CENTRE *for* CELL BIOLOGY
INSTITUTE OF CELL BIOLOGY, UNIVERSITY OF EDINBURGH**

MICHAEL SWANN BUILDING, KING'S BUILDINGS, MAX BORN CRESCENT, EDINBURGH EH9 3BF, SCOTLAND

William C. Earnshaw, Ph.D., FRS, FRSE, FMedSci

Professor and Wellcome Principal Research Fellow

phone +44 - (0)131 - 650-7101

FAX +44 - (0)131 - 650-7100

E-mail: bill.earnshaw@ed.ac.uk

**Chromosome
Structure
Lab**

July 5, 2024

Dr. Hironori Funabiki
Monitoring Editor
Journal of Cell Biology

Dan Simon, PhD
Scientific Editor
Journal of Cell Biology

Dear Hiro and Dan:

Thank you very much for overseeing the review process for our MS entitled "Near Millimolar Concentration of Nucleosomes in Human Mitotic Chromosomes from Late Prometaphase into Anaphase" with co-authors Fernanda Cisneros-Soberanis, Eva Simpson, Alison J. Beckett, Nina Pucekova, Samuel Corless, Natalia Y. Kochanova, Ian A. Prior, Daniel G. Booth and William C. Earnshaw. You will note that in responding to the helpful comments of the three referees, we have availed ourselves of the expertise of two additional laboratory members, and have therefore added two more authors to the MS (Corless & Kochanova).

We appreciate that two of the referees felt that our MS lacked novel mechanistic insights, but are very grateful to Referee 2 and to you for recognising that the structural information in the MS represents a tour de force that has potentially very significant implications for a field that is extremely active. To sum up our take-home message in one sentence - "How can SMC motors which have a stall force of 1 pN push huge chromatin loops in a forming mitotic chromosome where the concentration of nucleosomes exceeds 700 μm^{-3} ?" Whether the chromosomes are a separated nucleosome phase or gel - or simply extremely concentrated is something that others are looking at, and that goes beyond what we could hope to accomplish in this report.

Our realisation of the amazing concentration of nucleosomes in mitotic chromosomes and the fact that this concentration appears to plateau through central mitosis only became clear to us as we were writing the MS. We hope and believe that others interested in this problem will be as astounded as we were, and that this work will change the way biochemists think about mitotic chromosomes.

We have responded positively to all comments of the referees, making numerous changes to the MS (all revisions made in response to the referees are shown in blue font). In addition, in compliance with your suggestion, we have shortened the MS and converted it to Report format. We are aware that we have pushed the limits of what is allowed in a Report, but we did so in accordance with Dan's follow-up e-mail of July 1. Thus the document now has 27,664 characters in the Abstract, Introduction, Results & Discussion and Acknowledgements sections; in-text Figures have been cut to 5, and Supplementary figures have been increased to 4. There are 3 Supplementary Tables. We hope that our reading of your helpful e-mail was correct and that this is acceptable.

Specifically, we moved Figure 1, characterising the cell system, to the Supplement. We also revised Figure 5 (formerly 6) based on our efforts to respond to a comment by referee 1 by making our definition of the nucleosome parameters as rigorous as possible. Actually, we feel that analysing all published ATAC-seq data probably gives a state-of-the-art estimate of the percentage of nucleosome-free DNA in RPE-1 cells.

We hope that you will conclude that we have responded appropriately to all of the comments from yourselves and from the referees and that our reorganisation of the MS corresponds to what you had in mind. We would like to particularly express our gratitude to Hiro for again demonstrating the strength of having an active scientist as Monitoring Editor in a top scientific journal.

We will look forward to hearing your decision about the revised MS in due course.

Best regards,

Response to Reviewers:

Reviewer #1:

Advance Summary and Potential Significance to Field

In this manuscript, authors have used quantitative 3D electron microscopy to visualize and characterize changes to chromosome volume, surface area, width and length during mitotic progression between prometaphase through telophase in human RPE1 cells. The main conclusion is that chromosome volume reaches a plateau by prometaphase and chromosome volume remains constant during metaphase. Using the measured volume, they estimate a nucleosome concentration of ~666 micromolar, which is above the concentration for nucleosomes to undergo phase separation as determined in previous studies.

This is a nice, descriptive study of chromosome volume changes during mitosis. The manuscript is well written and the results are clearly presented. I enjoyed reading.

Comments to the Authors

My major comments are as follows:

1. The manuscript is very descriptive and does not offer new mechanistic insight into chromosome compaction/condensation during mitosis. It is not clear what are the mechanisms required to maintain the constant volume measured during metaphase or why this is functionally important.

We thank the reviewer for these comments and are happy they enjoyed reading our MS. It is true that this MS does not describe a mechanism for the compaction of mitotic chromatin. We and others have speculated on various mechanisms, including effects of chromatin modifications (Zhiteneva, PMID: 28903997) leading to possible phase separation (Rosen and Gerlich, PMID: 35922507). It is also widely believed that loop extrusion by condensin plays an important role and other factors may well be involved.

We do not attempt to describe the mechanism, but what we do set out to do is to define the limit concentration that is achieved, independent of mechanism. We conclude that when one does the calculation and looks at the resulting concentration, the result is extremely surprising. The amazingly high concentration that we determine here will significantly influence the sorts of detailed mechanisms and experimental systems that can be proposed in the future to understand the molecular mechanism of mitotic chromatin compaction. For example, can a motor with a stall force of 1 pN really push chromatin fibers (Samejima, PMID: 38659940; Prevo and Earnshaw, PMID: 38886215) into a phase where the concentration is in the hundreds of micromolar?

We should also point out that the existence of a plateau concentration from prometaphase into anaphase as measured here is novel and differs from results published by light microscopy. Indeed, a long standing issue with volume estimations using light microscopy data is the (comparatively) poor resolution in Z, which can result in inaccurate volume quantifications, typically retrieving over-

estimates. Our 3DCLEM analyses (both past and present) confirms this, as CLEM provides the utility to directly compare both LM and EM volume data sets, of the sample organelle.

Our result reveals that mitotic chromatin compaction may achieve a limit density that is even higher than that seen in phase-separated nucleosomes in vitro.

2. Some of the conclusions rely on the accuracy of volume and surface area measurements, however, as the authors demonstrate, these measurements (particularly surface area/granularity) are heavily influenced by the pixel size and resolution they pick.

The referee is correct that the sampling resolution (voxel size) has a substantial effect on surface area. However, in Figure S3 we present an analysis in which we systematically compare the effect of changing the voxel size while maintaining the same density threshold in a given dataset. We observed significant changes in the *surface area* after changing the voxel size. Importantly however, the *volume* remained nearly constant. Critically, all significant conclusions about the nucleosome concentration in the MS were based on volume measurements. We describe the changes in surface area during the different mitotic phases for a given voxel size, but our major conclusions do not depend on this.

3. Only a handful of chromosomes are measured in each phase of mitosis and all the major conclusions rely on these statistically low and underpowered datasets.

It is true that for Figures 1, 2 and 4 we show the entire group of chromosomes and only chromosome 1 on its own. However, using length, volume and primary constriction site (centromere position) we can confidently identify chromosomes 1-5 and 19-22 in all but early prometaphase and telophase cells. That is 18 chromosomes per cell, and we present an analysis of those chromosomes with respect to length, width and surface area individually in Figures 1-3 and S4.

Importantly, the major conclusion of the MS - determination of the total chromatin volume and calculation of the nucleosome density - was not based on results from single chromosomes, but instead from the sum total of ALL volumes for all chromosomes in each 3D reconstruction of an entire chromosome complement.

It is very unusual for a study to present volume EM generated 3D reconstructions for multiple cells in a single study. This is even more pronounced when 3DCLEM is required – to capture rare events (cells in specific stages of mitosis).

Briefly, the process to produce each individual reconstruction requires recognition of a suitable cell by light microscopy, relocating exactly the same cell after fixing, embedding and staining for electron microscopy (CLEM), positioning that cell at the center of a trimmed block face, typically the width of a single human hair (ie. very technically demanding), serial sectioning of the entire cell using the 3View microscope, density contouring to distinguish the chromosomes from the cytoplasm, and modelling in AMIRA to separate individual chromosomes from one another and, where possible, identify them. (This allowed us to compare various measurements with the known number of megabases in the given chromosome.) From start through analysis, in practice this process takes 2-3 months per cell if nothing goes wrong and there is no waiting list for access to the equipment. As a result, volume EM papers traditionally present a low number of samples (see Sajid (2021), Chen (2017), Booth (2016)), even those of Nobel Laureates (PMID 27110681).

The AMIRA modelling, particularly for anaphase cells (where chromosomes are closer together) is one of the greatest bottlenecks in this pipeline. Advances in segmentation tools, shifting towards fully automated “minute” modelling, are becoming more available, but are typically working with symmetrical cellular substructions or those with little overall deviation in conformation, such as vesicles and mitochondria. DGB has an ongoing collaboration with Amira software engineers, to generate AI and machine learning tools for rapid automated modelling of mitotic chromosomes, and can confirm that this technology is simply not sufficiently developed yet and thus, semi-automated (but highly accurate) segmentation is still the best approach.

Referee 2 was aware of this problem and complemented us on presenting complete reconstructions of 14 cells. We believe that this was critical, as, for example, independent reconstructions of 4 different metaphase cells revealed how reproducible the total chromosome volume is. Furthermore,

reconstructions of cells at different mitotic phases enabled us to examine the variation on volume across mitosis.

4. Major conclusions are speculative and depend on the accuracy of measurements (see comments 2 and 3). It is not clear what is the margin of error on the nucleosome concentration estimates, which is the main conclusion of the manuscript and the title of the manuscript. I think the title is not fully supported by the speculative, back of the envelope calculations based on a few chromosomes.

By terming our calculations "back of the envelope" the referee is not entirely accurate in describing the nature of our calculations.

Our calculation of the nucleosome concentration is based on hard measurements of the total chromatin volume and DNA content but ultimately must hinge on an assumption of the amount of DNA per nucleosome. Determination of the overall volume of all chromosomes in a cell is a hard number determined by our tomographic analysis. Using this total volume coupled with the golden path sequence of the human genome in our RPE1 clone, again yields a hard number of Mb/ μm^3 .

Where we must speculate is in specifying the number of base pairs per nucleosome. It is widely accepted that this number can vary in humans with 195 bp per nucleosome being commonly used for cultured cell lines (Tate and Philipson, PMID: 461205; Van Holde, 1988; Zou, PMID: 38749504).

We argue that it is in fact not possible to specify a single exact size (e.g. number of base pairs) for all nucleosomes in a human cell, as the spacing between nucleosomes varies in different regions of the chromatin and may well vary across the cell cycle. To try and address this issue rigorously we have examined data from 24 ATAC-seq datasets using ChIP-Atlas (Zou, PMID: 38749504). ATAC-seq identifies nucleosome-free regions in cellular chromatin. From this analysis, we find that only 0.3% of RPE1 DNA is likely to be nucleosome-free. The fraction of the RPE1 genome that is not in the ATAC-seq datasets corresponds to repetitive DNA sequences, which are almost certainly in heterochromatin and therefore likely to have regular nucleosome arrays. We conclude that our concentration of 760 μM (for metaphase chromosomes) is not simply "back of the envelope" but is instead the most accurate estimate that can be rigorously made at the present time.

We have added the following to the Methods (and added Sam Corless as an author to the MS):

lines 644 - 652:

An upper estimate of the proportion of RPE-1 DNA contained within nucleosome free regions was calculated from ATAC-seq "peak" data obtained from ChIP Atlas (track type class; ATAC-seq, cell type class; neuronal, threshold for significance; 200, cell type; hTERT RPE-1). Overlapping peaks were merged using 'bedtools merge', and the proportion of reads in 'peaks' (4.7Mbp) compared to the total length of non-repetitive DNA in the hg38 genome build (1.52Gbp), giving a value of 0.3% of the genome being localised to nucleosome depleted regions.

Reviewer #2 (Comments to the Authors (Required)):

Advance Summary and Potential Significance to Field

The manuscript by Fernanda Cisneros-Soberanis et al. presents a quantitative analysis of chromatin concentration measured during the progression through mitosis in RPE1 cells of human origin. In this study, the authors have further optimized their cell synchronization system, previously introduced in Gibcus et al. (2018), by implementing a presynchronization step. This step reduced the frequency of apoptosis and enabled the efficient synchronization of cells at specific mitotic stages, a feat not previously achieved to the best of my knowledge.

Additionally, correlative light and electron microscopy (CLEM) was employed to accurately determine the stage of mitosis for each cell. Subsequently, high-resolution data were acquired through serial block face scanning electron microscopy (SBF-SEM). Leveraging this advanced experimental setup, the authors investigated fundamental questions regarding chromatin compaction and decompaction during mitosis.

Their analysis unveiled an intriguing phenomenon: the density of mitotic chromatin remains remarkably constant and high from late prometaphase to early anaphase, which the authors term the "density plateau". This observation contradicts previous findings from light microscopy methods and offers essential novel insights into the organization and dynamics of mitotic chromatin.

SBF-SEM allows high-resolution data acquisition, but its throughput is much lower. Therefore, the considerable effort expended to record and reconstruct appreciable numbers of complete mitotic cells in 3D with impressive resolution, as shown in this study, should be appreciated.

The authors openly acknowledge the potential artifacts associated with the methods utilized, highlighting the necessity of cryo-electron volumetric imaging for overcoming these limitations. However, this technique is currently in its early developmental stage and cannot be applied. In their previous works, authors show that no discernable reduction of chromosome volumes occurs during sample preparation used in this work. Importantly, the key observation of the authors is based on a comparative analysis of different mitotic stages, which a possible minor structure cannot affect. The study combines a carefully organized series of experiments, resulting in valuable data. Results are described and supplemented with clear overview images and graphs. In summary the presented work is original and significant.

Suggested minor improvements

1. Lines 74-76. "They concluded that the maximal chromosome compaction occurs in late anaphase by an axial shortening of the chromosome arms. (Note, however, that the volume of a cylinder varies linearly with respect to length, but with the square of the radius.)" The reader would appreciate an explanation of this point. Does it mean that the radius measurements were not provided by Mora-Bermudez et al., therefore their statement is not conclusive?

Mora-Bermudez et al. conducted a detailed study on the chromosome volume in large scale from G2 to telophase, following the same cell using light microscopy (Resolution: 0.06 X 0.06 X 1.5 μm imaging every 2-5 min). From this experiment, they concluded that the maximal compaction is in late anaphase. Also, they selectively photoactivated a defined fragment of a chromosome during chromosome congression and followed it through the rest of mitosis (Resolution: 0.11 X 0.11 X 0.7-1.5 μm imaging every 0.5-4 min) and they concluded that this fragment shortened at the end of mitosis. However, they only measured the fragment length and density. The radius was not reported. Considering chromosomes as cylindrical structures, changes in volume are expected to increase linearly with length but with the square of the radius. Therefore, alterations in volume would likely be more significant if the radius is affected.

Lines 66-76 (original):

To investigate the kinetics of chromosome compaction during mitosis, the total chromatin volume was measured from G2 to telophase using live cell imaging in NRK epithelial cells (Mora-Bermudez et al, 2007). From interphase to mitosis, the chromatin volume decreased ~2- 3 fold. These cells constitutively expressed photoactivatable GFP-tagged H2B (H2B-PAGFP) allowing the authors to follow a defined segment of single chromosomes using photo-activation during chromosome congression and subsequent mitotic exit. They concluded that the maximal chromosome compaction occurs in late anaphase by an axial shortening of the chromosome arms. (Note, however, that the volume of a cylinder varies linearly with respect to length, but with the square of the radius.)

Lines 68-70 (revised):

To investigate the kinetics of chromosome compaction during mitosis, the total chromatin volume was measured from G2 to telophase using live cell imaging in NRK epithelial cells (Mora-Bermudez et al, 2007). From interphase to mitosis, the chromatin volume decreased ~2- 3 fold. These cells constitutively expressed photoactivatable GFP-tagged H2B (H2B-PAGFP) allowing the authors to follow a defined segment of single chromosomes using photo-activation during chromosome congression and subsequent mitotic exit. They concluded that maximal chromosome compaction occurs in late anaphase due to an axial shortening of the chromosome arms. However, the volume of a cylinder increases linearly with length but with the square of the radius, and the width of the chromatids was not reported.

2. Lines 186: "This technique allows us to achieve a resolution in X and Y of 4 nm..." Do you mean sampling step? The resolution of SEM is not directly linked to the pixel size, but depends of the electron optics quality, voltage, and likely the detector as you mention later.

We appreciate the observation of the referee. When we refer to a resolution in X and Y of 4 nm, we indeed mean the sampling step rather than the intrinsic resolution of the SEM. As you correctly pointed out, the true resolution in the SEM is influenced by factors such as the electron optics quality, the voltage applied, and the detector efficiency. Therefore, while the sampling step is 4 nm, the actual resolution would be determined by these additional parameters.

Lines 186 (original):

This technique allows us to achieve a resolution in X and Y of 4 nm

Line 157 (revised):

Scanning Electron Microscopy (SBF-SEM) with a sampling resolution of 4

3. Lines 238-240 "This ~20% difference in volume is likely due to the combination of improved resolution of a new detector in the 3View system..." This statement can be supported by providing resolution measurements for the previous and current detector performed on any standard sample. This can also be illustrated by showing the difference in density profile between the previous and current data (as shown in Figure 2A). The lower resolution will be evident in the smoother transition of gray values between the cytoplasm and chromosome.

We appreciate the referee's suggestion to clarify the text. The sampling frequency for our current data sets was between 4-8 nm, and varies depending on the cell, whereas the sampling frequency for the metaphase cell in Booth et al 2016 was 24 nm in X and Y (further binned 2x2). Supplementary Figure 3 shows that binning the image causes an increase in the measured volume. We include for the Editor a figure similar to Figure 2A to illustrate the differences between the studies. We observed a similar transition between the chromosomes and the cytoplasm. We also note that in the older study, there was a higher contrast in the staining, which could also affect the definition of the chromosome edge. Furthermore, the previous study did not measure the separation between the spindle poles, so the cell could have in fact been in late prometaphase, where the volume is greater.

Lines 236-240 (original):

In a previous study, we found the volume of a single human metaphase plate to be $176.9 \mu\text{m}^3$ (Booth et al, 2016). This ~20% difference in volume is likely due to the combination of improved resolution of a new detector in the 3View system and the use of a different RPE1 hTERT cell clone in those experiments.

Lines 203-207 (revised):

In a previous study, we found the volume of a single human metaphase plate to be $176.9 \mu\text{m}^3$ (Booth et al, 2016). This ~20% difference in volume is likely due to the combination of improved sampling resolution of a new detector in the 3View system (previously 24 nm in X and Y, currently 4 nm in X and Y), image binning in the previous study (see Fig S3), the use of the RPE1 hTERT cell clone in those experiments, the precise mitotic stage, and differences in the uranyl acetate staining.

4. Lines 278-283. The observation of the evolution of the shape of chromosome or chromatid profiles during mitosis is very interesting. Having a figure illustrating this interpretation and discussion would be useful for a better understanding, especially because the current study investigates more phases of mitosis than the previous one. Could it be compatible with and/or support the compaction hypothesis based on phase separation?

We were not at first certain exactly what the referee was asking for.

Lines 276-283 state:

Notably, the width of the chromosome is about 10% less than the sum of the widths of the two sister chromatids. This suggests three possibilities. First, the sister chromatids may be compressed along their interface with one another. Second, there may be a roughly 0.06 μm overlap (mixing) of the two sister chromatids. Third, chromatin could be "squeezed" out of the plane of pairing (i.e. pushed above or below the plane of contact of the two sisters). Our data argue against possibility 3, as when viewed in cross section the sister chromatids display a circular profile, not an ellipsoid as model 3 would suggest.

The issue of apparent overlap between the two cohesed sister chromatids is complex, and one that we feel cannot be definitively answered using the methods available for the present study. We believe that to do this would require high-resolution imaging of the individual sister chromatids *in situ* during metaphase in order to demonstrate whether they were indeed mixed in the cohesed region (as might be expected for simple phase separation) or whether the individual sisters remain separate (as might be expected if chromosomes were shaped by a scaffold). Attempts to examine this question using EdU staining in another study (in DT40 cells, where the sister cohesion is more pronounced than it is in RPE-1) were inconclusive, and we feel that this should remain a question for a future study. We cannot exclude, for example, that chromosomes are shaped by a non-histone scaffold, but that around that scaffold they are concentrated by phase separation. We would therefore prefer not to include a speculative figure at this time, particularly since this is not critical for our main message, which is the high plateau concentration of nucleosomes in mid mitosis.

5. Do you see only surface granularity in prometaphase chromosomes, or are there also internal structural irregularities - cavities that have been demonstrated in some cell lines? (for example in Schroeder-Reiter E., et al. J Struct Biol. 2009;165:97-106).

We do not claim that chromatin in mitotic chromosomes is uniform, but we did not observe internal cavities as reported by Schroeder-Reiter et al. (2009) in isolated barley chromosomes. However, some studies on isolated human chromosomes using SEM or ET did detect them (Shemilt, PMID: 24470422; Phengchat, PMID: 31539626; Sumner, PMID: 1893796). They were also seen in a study (Chen, PMID: 28776025) in which human cells were hypotonically swollen and "fixed" with methanol-acetic acid (which is an *in situ* precipitation and not a true fixation procedure). We speculate that the cavities seen in all of those studies could result from sample preparation procedures. Importantly, it has been reported that chromosome structure is significantly affected by the solutions or buffers used during preparation (Sone, PMID: 12680460, plus many unpublished results from our own work).

Other studies describing chromosome structure by electron microscopy within cells did not report observing those cavities (Heslop-Harrison, PMID: 2606474; Sajid, PMID: 34206020; Booth, PMID: 27840028; Ou, PMID: 28751582). Nonetheless, these studies also reported that the chromatin is not uniform. Although it is tempting to analyze the chromatin density in our images, we believe that the chemical fixation used to process the RPE1 cells could potentially introduce artifacts at this level. For future studies, it will be valuable to evaluate chromatin density within chromosomes using high-pressure freezing to vitrify the cells, which would be expected to yield more physiological results.

6. Lines 340-354. You talk about the characteristics of the largest chromosome in prometaphase, and then move on to chromosome 1 in the context of the prometaphase-metaphase transition (Line 351). This sounds confusing. I understood that unambiguous chromosome recognition was not possible in prometaphase? Could you please clarify this?

We thank the referee for pointing this out. Indeed, particular individual chromosomes could not be identified in early prometaphase (Prometaphase 1 and 2) because our identification algorithm depends on both the size of the chromosome and the position of the centromere - defined by the primary constriction. In these early prometaphases, we could not resolve clear centromeric constrictions, which became apparent only in late prometaphase (Prometaphase 3 and 4). We have attempted to clarify the text.

Lines 340-354 (original):

Another difference between earlier and later prometaphase was seen in the granularity of the chromosome surface, which decreased as prometaphase progressed. As prometaphase progresses, the chromosome volume and surface decrease (Fig 3D). The volume for the largest chromosome decreased by 37% from 10.03 μm^3 in

the earliest prometaphase (as defined by the spindle pole-to-pole spacing) to 6.35 μm^3 in metaphase. The corresponding reduction in surface area for the same chromosome was from 107.28 μm^2 to 44.95 μm^2 in metaphase, a reduction of 58%. Thus, as prometaphase progresses, the chromosome volume is reduced, but the surface area (a measure of granularity) decreases even more dramatically.

The length of chromosome 1 also decreased by almost 50% between the earliest prometaphase and metaphase (from 13.4 μm to 7.3 μm). Indeed, shortening of the chromosomes appeared to be an ongoing gradual process during prometaphase (Fig 3E).

Line 294 (revised):

Another difference between earlier and later prometaphase was seen in the granularity of the chromosome surface, which decreased as prometaphase progressed. As prometaphase progresses, the chromosome volume and surface decrease (Fig 3D). The volume for the largest chromosome decreased by 37% from 10.03 μm^3 in the earliest prometaphase (as defined by the spindle pole-to-pole spacing) to 6.35 μm^3 in metaphase. The corresponding reduction in surface area for the same chromosome was from 107.28 μm^2 to 44.95 μm^2 in metaphase, a reduction of 58%. Thus, as prometaphase progresses, the chromosome volume is reduced, but the surface area (a measure of granularity) decreases even more dramatically.

The length of the largest chromosome also decreased by almost 50% between the earliest prometaphase and metaphase (from 13.4 μm to 7.3 μm). Indeed, shortening of the chromosomes appeared to be an ongoing gradual process during prometaphase (Fig 3E).

7. Lines 355-358. "Thus, the linear density of chromatin appears to remain relatively constant as chromatids shorten during early mitosis." As I understand it, linear density is determined solely by chromatid length (Mb/ μm). How can it remain constant if chromatids become shorter, or am I misunderstanding?

The referee raises an excellent point. We considered "linear density" the DNA content per unit of chromosome length (μm). The conundrum occurs because we use the chromosome length per chromosome volume ($\mu\text{m}/\mu\text{m}^3$) to describe changes in the chromosome length relative to the chromosome volume. In prometaphase, as the chromosomes were shortening, they were also undergoing a proportional compaction. We have re-worded the text to try and avoid this confusion.

Lines 354-358 (original):

Surprisingly, analysis of the relationship between the total chromosome length and total chromosome volume revealed a relatively constant ratio of $1.3 \pm 0.04 \mu\text{m}/\mu\text{m}^3$. Thus, the **linear density** of chromatin appears to remain relatively constant as chromatids shorten during early mitosis.

Lines 301-304 (revised):

Surprisingly, analysis of the relationship between the total chromosome length and total chromosome volume revealed a relatively constant ratio of $1.3 \pm 0.04 \mu\text{m}/\mu\text{m}^3$. Thus, chromatin compaction appears to increase in proportion to chromatid shortening during early mitosis.

8. Samejima et al, 2024, missing citation.

We have provided the relevant citation (the MS was not yet in bioRxiv at time of our submission).

Samejima, K., Gibcus, J. H., Abraham, S., Cisneros-Soberanis, F., Samejima, I., Beckett, A. J., . . . Earnshaw, W. C. (2024). Rules of engagement for condensins and cohesins guide mitotic chromosome formation. *bioRxiv*, 2024.2004.2018.590027. doi:10.1101/2024.04.18.590027

9. Statistical analysis. Using non-parametric tests like the Kolmogorov-Smirnov test would be more suitable for comparing relatively small distributions, especially when their normality cannot be justified.

We thank the referee for this suggestion. Before conducting a t-test to determine any significant differences, we carried out an assessment of our data's distribution using the Shapiro-Wilk test. Our data showed a p-value > 0.05 , indicating a normal distribution. The details of this analysis will be included in the statistical analysis section within the Materials and Methods. These modifications have been made in consultation with Dr. Natalia Kochanova, who has been added as an author of the MS.

Lines 714-717 (original):

Statistical analysis.

Chromosome width in the different mitotic stages was analysed using a 2-tail t-student test from the average of 10 measurements. Linear regressions were performed using R and analysed using ANOVA test.

Lines 637-641 (revised):

Statistical analysis.

For chromosome/chromatid width analysis, we averaged 10 measurements per chromosome per cell. We assessed the normality of the data using the Shapiro-Wilk test. Our data showed a p-value > 0.05, indicating a normal distribution. Then, we conducted a 2-tail t-student test to determine any significant differences. Linear regressions were performed using R and analysed using an ANOVA test.

However, we also followed the suggestion of the referee and re-analysed the data using a Kolmogorov-Smirnov test. The resulting p-values are shown in the table (included here for the referee).

	t-test	KS test
Metaphase chromosome width (Fig 2G up)	0.1915	0.3315
Metaphase chromatid width (Fig 2G down)	0.0044	0.078*
Anaphase chromatid width (Fig 4G)	0.0049	0.007

Reviewer #3 (Comments to the Authors (Required)):

Advance Summary and Potential Significance to Field

Summary: Cisneros-Soberanis et al. present a detailed quantitative analysis of chromosome compaction, which is essential for accurate chromosome segregation during mitosis, using Volume Electron Microscopy. Their research maps chromosomes with nanometer precision from prometaphase to telophase in human RPE1 cells. Notably, during prometaphase, the chromosomes exhibit primary constriction, leading to a smoother surface and shortened arms. The study highlights significant progressive compaction by late prometaphase that remains consistent through metaphase and early anaphase. Additionally, it is observed that chromosome volume increases just before the nuclear envelope reformation in late anaphase. These results support the proposed chromatin condensation in mitosis reaches and maintains a constant density threshold from late prometaphase through early anaphase.

Contribution to Mitotic Chromosome Formation Studies: This study supports previous theories proposed by Cremer (2003), Sajid (2021), and Beel (2021) concerning the importance of size and density in the formation and positioning of mitotic chromosomes. Although the study does not introduce novel information, it provides an intriguing speculation that the observed density limits during chromosome compaction could be related to phase separation. This hypothesis gains importance considering its innovative potential in the field, which has been further explored in recent studies by Gibson (2023 and 2019).

Comments to the Authors

Overall, the study does not add novelty to the field at this publication stage. However, the field is open to many questions, encouraging the authors to pursue these lines of inquiry further in the study.

This lack of novelty is the major criticism of this referee.

However, in considering the work cited above we point out that:

- 1- The work of Cremer et al. used light microscopy to examine chromosome positioning in interphase in different cell lines. This is quite a different question from our study.
- 2- The study from Sajid et al used SBF-SEM to describe a single human late prophase cell following hypotonic swelling. This was pioneering work, but they did not go beyond describing the position and distribution of the chromosomes in the prophase nucleus, and since they used hypotonic swelling, they could not accurately measure the volume of the chromosomes.

-3- The study from Beel and Kornberg used cryo-ET of chromosomes to look at the fine structure of the nucleosome fiber in chromosomes purified after hypotonic swelling. Thus, they did not look at the organisation of mitotic chromatin in intact cells.

None of these studies performed a 3D reconstruction of cells to determine structural parameters of chromosomes in cells at different stages of mitosis. Thus, none of them was able to demonstrate or even suggest the principal observation of our work - that mitotic chromatin appears to achieve a "plateau density" during metaphase and the surrounding prometaphase and anaphase.

We conclude that in contrast to the view of the referee, our report does indeed introduce "novel" information.

Potential for Broader Impact through Comparative Analysis:

In the following, the referee goes on to suggest a number of ways in which we could expand our analysis. Although we agree that a number of these might yield interesting new information, in general these suggestions all propose studies that are beyond the scope of the present analysis.

This study could significantly benefit from extending its analysis to compare stable karyotypical mammalian endothelial RPE1 cells with cancer cells. This comparison aims to identify variations in chromosome structure and the stability of nucleosome concentrations during mitosis.

The goal of our study was to establish the base-line parameters for chromatin packing in mitotic chromosomes in normal human cultured cells. This is one of the most basic features of mitotic chromosomes, but yet has essentially remained elusive (due to technical limitations) for over 100 years. It could indeed be interesting to examine the detailed structure and organisation of chromosomes in cancer cells, but the data would be very difficult to interpret quantitatively as we have done here, due to the existence of aneuploidy and highly variable translocations between chromosomes of the cancer cells which would likely vary within the population. As a result 3D microscopy might yield interesting information about chromosome behaviour, but would likely not be suitable for the questions about nucleosome packing density that we have asked here.

The use of RPE1 hTERT CDK1as cells as the model is well-justified, providing an advantage over alternatives like nocodazole. However, introducing motor protein inhibitors such as CENP-E or even exploring the role of SMC motors in DNA extrusion could offer novel insights into the forces and error correction mechanisms that influence chromosome structure during mitotic compaction.

Dr. Cisneros-Soberanis and a student have contributed to a second paper describing the results of a 7 year collaboration with Job Dekker, Leonid Mirny and Anton Goloborodko that is currently under revision at another journal. That study contains 3D structures of chicken DT40 cells lacking cohesin, condensin I and condensin II. Interestingly, the chicken chromosomes (both wild type and mutant) also have a very highly conserved ultimate chromatin packing density in mid-mitosis which differs from that in human cells, as mentioned in our Results and Discussion. So, the other study shows that this approach can be used to examine the structure of mutant chromosomes (in that study, all are in late prometaphase). However, we believe that the present study is important in setting a baseline for the structural parameters of normal chromosomes during unperturbed mitotic progression in human cells.

Further Research Considerations: Including a control group of mitotic cells that are not synchronised could shed light on whether the slower progression into mitosis of the RPE1 hTERT CDK1as cells impacts chromosome compaction.

In studies of DT40 cells we have examined the effects of various synchronisations and depletion of differing combinations of SMC complexes in the timing of mitotic entry and chromosome morphology at the light microscope level. Any changes were very subtle and we are unsure whether EM would be enough sensitive to detect such subtle changes. Thus, this is an interesting suggestion that we believe is beyond the scope of the present study.

Methodological Enhancements: It would be beneficial to display the flow cytometry DNA content distribution within the cell cycle in Figure 1C and provide clear examples to delineate each phase in Figure 1D.

We do not have flow cytometry data for the experiment shown in Figure 1C (current Figure S1C). Instead, interphase, mitotic and apoptotic cells were counted using the LM and DAPI staining. 1000 cells were counted per condition. We believe that this provides a suitable level of accuracy.

We are not certain whether the referee is asking us to include light microscopy images in Figure 1D (current Figure S1D). This could be done in a supplementary figure, but we question the novelty and utility of such figures, as there are many examples of the mitotic stages of RPE1 cells in the literature and in fact, with the exception of prophase, we have provided 2D reconstructions of cells at the various mitotic stages that go far beyond what we could achieve by light microscopy.

Additionally, the authors should present the nucleosome imagery used for quantification alongside a 3D tomographic visualisation of each mitotic phase to enhance the understanding and reproducibility of their findings.

We are not certain what is being requested here. Our imaging modality did not have the resolution to see individual nucleosomes in mitotic cells (something that is at or beyond state-of-the-art even with high voltage cryoElectron tomography). The tomograms (reconstructions of entire cells) used for calculation of the nucleosome density are in fact the tomograms shown in Figures 1, 2 and 4 and the reproducibility of the limit nucleosome density is shown in Figure 5.

July 23, 2024

RE: JCB Manuscript #202403165R

Prof. William C Earnshaw
University of Edinburgh
Institute of Cell Biology
University of Edinburgh, Swann Building, Kings Buildings
Max Born Crescent
Edinburgh EH9 3BF
United Kingdom

Dear Prof. Earnshaw,

Thank you for submitting your revised manuscript entitled "Near Millimolar Concentration of Nucleosomes in Human Mitotic Chromosomes from Late Prometaphase into Anaphase." We would be happy to publish your paper in JCB pending final revisions necessary to meet our formatting guidelines (see details below).

A. MANUSCRIPT ORGANIZATION AND FORMATTING:

1) Figure formatting: Reports may have up to 5 main text figures. Scale bars must be present on all microscopy images, including inset magnifications. Molecular weight or nucleic acid size markers must be included on all gel electrophoresis. Please add scale bars to Figures 1A/C, 3B, & S1E and MW markers to S1A.

Also, please avoid pairing red and green for images and graphs to ensure legibility for color-blind readers. If red and green are paired for images, please ensure that the particular red and green hues used in micrographs are distinctive with any of the colorblind types. If not, please modify colors accordingly or provide separate images of the individual channels.

2) Statistical analysis: Error bars on graphic representations of numerical data must be clearly described in the figure legend. The number of independent data points (n) represented in a graph must be indicated in the legend. Please, indicate whether 'n' refers to technical or biological replicates (i.e. number of analyzed cells, samples or animals, number of independent experiments). If independent experiments with multiple biological replicates have been performed, we recommend using distribution-reproducibility SuperPlots (please see Lord et al., JCB 2020) to better display the distribution of the entire dataset, and report statistics (such as means, error bars, and P values) that address the reproducibility of the findings.

Statistical methods should be explained in full in the materials and methods. For figures presenting pooled data the statistical measure should be defined in the figure legends. Please also be sure to indicate the statistical tests used in each of your experiments (both in the figure legend itself and in a separate methods section) as well as the parameters of the test (for example, if you ran a t-test, please indicate if it was one- or two-sided, etc.). Also, if you used parametric tests, please indicate if the data distribution was tested for normality (and if so, how). If not, you must state something to the effect that "Data distribution was assumed to be normal but this was not formally tested."

3) Title: To enhance accessibility for a broad cell biology audience JCB discourages the usage of species names in titles. We therefore recommend removing "human" from your title.

4) Materials and methods: Should be comprehensive and not simply reference a previous publication for details on how an experiment was performed. Please provide full descriptions (at least in brief) in the text for readers who may not have access to referenced manuscripts. The text should not refer to methods "...as previously described."

5) For all cell lines, vectors, constructs/cDNAs, etc. - all genetic material: please include database / vendor ID (e.g., Addgene, ATCC, etc.) or if unavailable, please briefly describe their basic genetic features, even if described in other published work or gifted to you by other investigators (and provide references where appropriate). Please be sure to provide the sequences for all of your oligos: primers, si/shRNA, RNAi, gRNAs, etc. in the materials and methods. You must also indicate in the methods the source, species, and catalog numbers/vendor identifiers (where appropriate) for all of your antibodies, including secondary. If antibodies are not commercial, please add a reference citation if possible.

6) Microscope image acquisition: The following information must be provided about the acquisition and processing of images:

- Make and model of microscope
- Type, magnification, and numerical aperture of the objective lenses
- Temperature
- Imaging medium
- Fluorochromes
- Camera make and model
- Acquisition software
- Any software used for image processing subsequent to data acquisition. Please include details and types of operations involved (e.g., type of deconvolution, 3D reconstitutions, surface or volume rendering, gamma adjustments, etc.).

7) References: There is no limit to the number of references cited in a manuscript. References should be cited parenthetically in the text by author and year of publication. Abbreviate the names of journals according to PubMed.

8) Supplemental materials: Please also note that tables, like figures, should be provided as individual, editable files. A summary of all supplemental material should appear at the end of the Materials and methods section. Please include one brief sentence per item.

9) eTOC summary: A ~40-50 word summary that describes the context and significance of the findings for a general readership should be included on the title page. The statement should be written in the present tense and refer to the work in the third person. It should begin with "First author name(s) et al..." to match our preferred style.

10) Conflict of interest statement: JCB requires inclusion of a statement in the acknowledgements regarding competing financial interests. If no competing financial interests exist, please include the following statement: "The authors declare no competing financial interests." If competing interests are declared, please follow your statement of these competing interests with the following statement: "The authors declare no further competing financial interests."

11) A separate author contribution section is required following the Acknowledgments in all research manuscripts. All authors should be mentioned and designated by their first and middle initials and full surnames. We encourage use of the CRediT nomenclature (<https://casrai.org/credit/>).

12) ORCID IDs: ORCID IDs are unique identifiers allowing researchers to create a record of their various scholarly contributions in a single place. Please note that ORCID IDs are required for all authors. At resubmission of your final files, please be sure to provide your ORCID ID and those of all co-authors.

13) JCB requires authors to submit Source Data used to generate figures containing gels and Western blots with all revised manuscripts. This Source Data consists of fully uncropped and unprocessed images for each gel/blot displayed in the main and supplemental figures. Since your paper includes cropped gel and/or blot images, please be sure to provide one Source Data file for each figure that contains gels and/or blots along with your revised manuscript files. File names for Source Data figures should be alphanumeric without any spaces or special characters (i.e., SourceDataF#, where F# refers to the associated main figure number or SourceDataFS# for those associated with Supplementary figures). The lanes of the gels/blots should be labeled as they are in the associated figure, the place where cropping was applied should be marked (with a box), and molecular weight/size standards should be labeled wherever possible. Source Data files will be directly linked to specific figures in the published article.

**** Please add Source Data images for blots in Figure S1A. ****

14) Journal of Cell Biology now requires a data availability statement for all research article submissions. These statements will be published in the article directly above the Acknowledgments. The statement should address all data underlying the research presented in the manuscript. Please visit the JCB instructions for authors for guidelines and examples of statements at (<https://rupress.org/jcb/pages/editorial-policies#data-availability-statement>).

B. FINAL FILES:

-- High-resolution figure and MP4 video files: See our detailed guidelines for preparing your production-ready images,

<https://jcb.rupress.org/fig-vid-guidelines>.

Thank you for your attention to these final processing requirements. Please revise and format the manuscript and upload materials within 7 days. If you need an extension for whatever reason, please let us know and we can work with you to determine a suitable revision period.

Thank you for this interesting contribution, we look forward to publishing your paper in Journal of Cell Biology.

Sincerely,

Hironori Funabiki, PhD
Monitoring Editor
Journal of Cell Biology

Dan Simon, PhD
Scientific Editor
Journal of Cell Biology

Reviewer #2 (Comments to the Authors (Required)):

I am fully satisfied with the responses and corrections made by the authors. I also appreciate the authors efforts to reorganize the manuscript according to the report format.

The extremely high concentration of chromatin in mitosis reported by the authors is indeed a very fundamental message that I am sure will be appreciated by the chromatin community.

I fully support the publication of the manuscript.